# Bandwidth-tuned Mott transition and superconductivity in moiré WSe$_2$

Yiyu Xia[1,6 ✉], Zhongdong Han[2,3,6 ✉], Jiacheng Zhu[1], Yichi Zhang[2], Patrick Knüppel[2], Kenji Watanabe[4], Takashi Taniguchi[4], Kin Fai Mak[1,2,3,5 ✉] & Jie Shan[1,2,3,5 ✉]

The emergence of high-transition-temperature ($T_c$) superconductivity in strongly correlated materials remains the main unsolved problem in physics. High-$T_c$ materials, such as cuprates, are generally complex and not easily tunable, making theoretical modelling difficult. Although the Hubbard model—a simple theoretical model of interacting electrons on a lattice—is believed to capture the essential physics of high-$T_c$ materials[1–5], obtaining accurate solutions of the model, especially in the relevant regime of moderate correlation, is challenging[6]. The recent demonstration of robust superconductivity in moiré WSe$_2$ (refs. 7,8), in which low-energy electronic bands can be described by the Hubbard model and are highly tunable[9–11], presents a new platform for studying the high-$T_c$ problem. Here we tune moiré WSe$_2$ bilayers to the moderate correlation regime through the twist angle and map the phase diagram around one hole per moiré unit cell ($v = 1$) by electrostatic gating and electrical transport and magneto-optical measurements. We observe a range of high-$T_c$ phenomenology, including an antiferromagnetic insulator at $v = 1$, superconducting domes on electron and hole doping, and unusual metallic states such as strange metals[12–14]. Twist-angle dependence studies further show that the highest $T_c$ always occurs adjacent to the Mott transition[3,15]. Our results indicate strong correlation as the key to superconductivity in moiré WSe$_2$ and establish a new material system for studying high-$T_c$ superconductivity in a controllable manner.

The Hubbard model, describing electrons hopping on a lattice with a hopping amplitude $t$ between neighbouring sites and an on-site electron–electron repulsion $U$, provides a simplified representation of high-temperature superconductors[1–5]. The electron hopping leads to a finite bandwidth $W$ (=$8t$ and $9t$, respectively, for square and triangular lattices). In cuprates, $W$ and $U$ are comparable[3,5], indicating that these materials are in the moderate correlation regime and are near a Mott transition from an insulator to a metal[3,15]. The idea of doping a Mott insulator for superconductivity has been extensively studied for decades[1–5]. Although the nature of the superconducting state itself has been qualitatively understood, a full understanding of the rich phase diagram remains unknown because accurately solving the Hubbard model in the moderate correlation regime, in which different orders intricately compete in the ground state, is difficult[6]. Studying high-transition-temperature ($T_c$) phenomenology in a tunable quantum system[16,17] may, therefore, shed light on how superconductivity emerges and pave the way for designing new high-$T_c$ materials.

Transition metal dichalcogenide (TMD) moiré heterobilayers, such as WSe$_2$/WS$_2$, are established quantum simulators of the two-dimensional (2D) triangular lattice Hubbard model[18,19]. For small twist angles, the moiré period (consequently $U/W$) is largely determined by the lattice mismatch in heterobilayers. Studies so far have focused on the strong correlation limit, and superconductivity has not yet been realized. The recent demonstration of robust superconductivity in twisted WSe$_2$ (tWSe$_2$) homobilayers[7,8], for which $U/W$ can be readily tuned by the twist angle[20–22], provides the possibility of exploring the superconducting phase diagram in the moderate correlation regime[23–25]. Here, we report the phase diagram of tWSe$_2$, focusing on twist angle around 4.6°, for which $U$ and $W$ are comparable. The transport characteristics, supplemented by magnetic susceptibility, reveal rich high-$T_c$ phenomenology, including an antiferromagnetic (AF) insulator, superconducting domes and strange metals. The phase diagrams for both electron and hole doping of the AF insulator and their continuous evolution as the system undergoes a band-structure-tuned Mott transition can be obtained using a single dual-gated device (Fig. 1a).

## Twist angle effects

Monolayer WSe$_2$ is a triangular lattice semiconductor with its valence band maxima located at the K and K′ points of the hexagonal Brillouin zone (BZ)[19]. A triangular moiré lattice with period $a_M = \frac{a}{\sqrt{2(1 - \cos\theta)}}$ is formed when two layers are stacked with a relative twist angle $\theta$, where $a = 0.33$ nm is the monolayer lattice constant. Figure 1b,c shows, respectively, the band structure and density of states (DOS) as a function of hole filling factor ($v$) and vertical electric field ($E$) for 4.6° tWSe$_2$ from the continuum model calculations (Methods). The K-valley states

[1]School of Applied and Engineering Physics, Cornell University, Ithaca, NY, USA. [2]Laboratory of Atomic and Solid State Physics, Cornell University, Ithaca, NY, USA. [3]Max Planck Institute for the Structure and Dynamics of Matter, Hamburg, Germany. [4]National Institute for Materials Science, Tsukuba, Japan. [5]Kavli Institute at Cornell for Nanoscale Science, Ithaca, NY, USA. [6]These authors contributed equally: Yiyu Xia, Zhongdong Han. ✉e-mail: yx579@cornell.edu; zh352@cornell.edu; kin-fai.mak@mpsd.mpg.de; jie.shan@mpsd.mpg.de

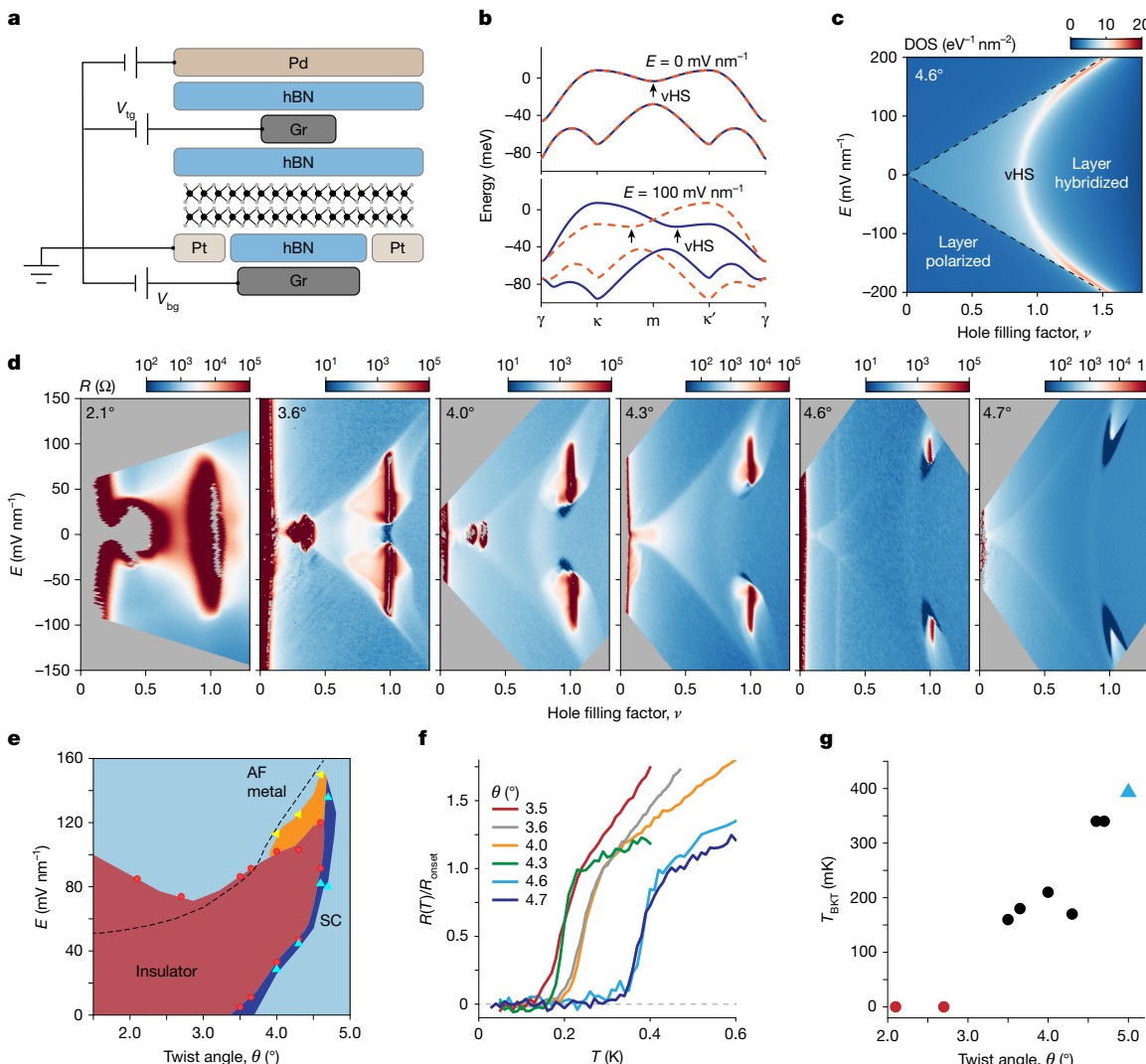

**Fig. 1 | Twist angle effects. a**, Schematic side view of dual-gated transport devices. The tWSe₂ sample is contacted by platinum (Pt) electrodes and controlled by voltages $V_{tg}$ and $V_{bg}$ applied on the hBN/Gr gates. The Pd contact and split gates turn on the Pt contacts and turn off the parallel channels, respectively. **b**, Topmost moiré valence bands for the K-valley state (blue solid line) and K′-valley state (orange dashed line) at $E = 0$ mV nm⁻¹ (top) and 100 mV nm⁻¹ (bottom). Arrows mark the vHS. **c**, Electronic DOS as a function of $\nu$ and $E$. The vHS with high DOS shifts towards higher $\nu$ with increasing $E$. Results in **b** and **c** are from the continuum model calculations for 4.6° tWSe₂. **d**, Resistance

as a function of $\nu$ and $E$ at 50 mK in tWSe₂ with twist angle varying from 2.1° to 4.7°. **e**, $\theta$–$E$ phase diagram at $\nu = 1$ constructed from experiment (symbols). Red, correlated insulator; dark blue, superconductor (SC); yellow, AF metal; light blue, metal. **f**, Superconducting transitions (for the highest $T_c$) for different twist angles. **g**, Highest $T_{BKT}$ (Berezinskii–Kosterlitz–Thouless transition temperature) versus $\theta$. The blue triangle is from ref. 8. Dashed lines in **c** and **e** denote the boundary between the layer-hybridized and layer-polarized regions.

of the top and bottom layers fold, respectively, onto the κ and κ′ valleys of the moiré BZ, which are swapped for the K′-valley states[9–11,26,27]. The bands from the two layers hybridize and generate a saddle point at which they intersect. The hybridized bands remain spin degenerate for small $\theta$ because of spin–valley locking in monolayer TMDs[19] (Fig. 1b, top). Application of a finite $E$-field places the two layers at different potentials and lifts the spin degeneracy[10] (Fig. 1b, bottom); it also shifts the van Hove singularity (vHS) with a diverging DOS continuously from $\nu < 1$ to $\nu > 1$ (Fig. 1c). Sufficiently large fields eventually polarize the holes to one of the layers. Theoretical studies showed that the topmost moiré valence band, if nontopological, can be described by the triangular lattice Hubbard model with an additional $E$-dependent spin–orbit coupling term[9–11]. This applies to samples with $\theta \geq 4°$ (ref. 28).

Figure 1d shows resistance $R$ as a function of $E$ and $\nu$ at temperature $T = 50$ mK in a series of dual-gated tWSe₂ devices with twist angle increasing from 2.1° to 4.7° (see Methods for details on the device

fabrication and characterizations; see Extended Data Fig. 1 for a device image; see Extended Data Fig. 2 for the determination of the moiré density). The resistance maps qualitatively agree with the DOS map in Fig. 1c, including a layer-hybridized region centred at $E = 0$ and a vHS with enhanced resistance due to the large DOS. Not captured by the band theory are the insulating states at fractional fillings of the first moiré band, $\nu = 1$, 1/3 and 1/4, which exhibit strong electron correlations. As the twist angle increases, the correlated insulators gradually melt and are indiscernible at 4.7°. This is expected because the correlation effects weaken as the moiré period decreases. The $E$-field dependence of the $\nu = 1$ state is further enriched by the presence of the vHS, which enhances the correlation effects[7,8,10,20,29] (see Methods for further discussions).

Superconductivity starts to emerge near $\theta = 3.5$–3.6° and persists over a range of twist angle up to about 5° (beyond which superconductivity is expected to fade away together with the moiré effect). Rather than following the vHS, the most robust superconducting state always

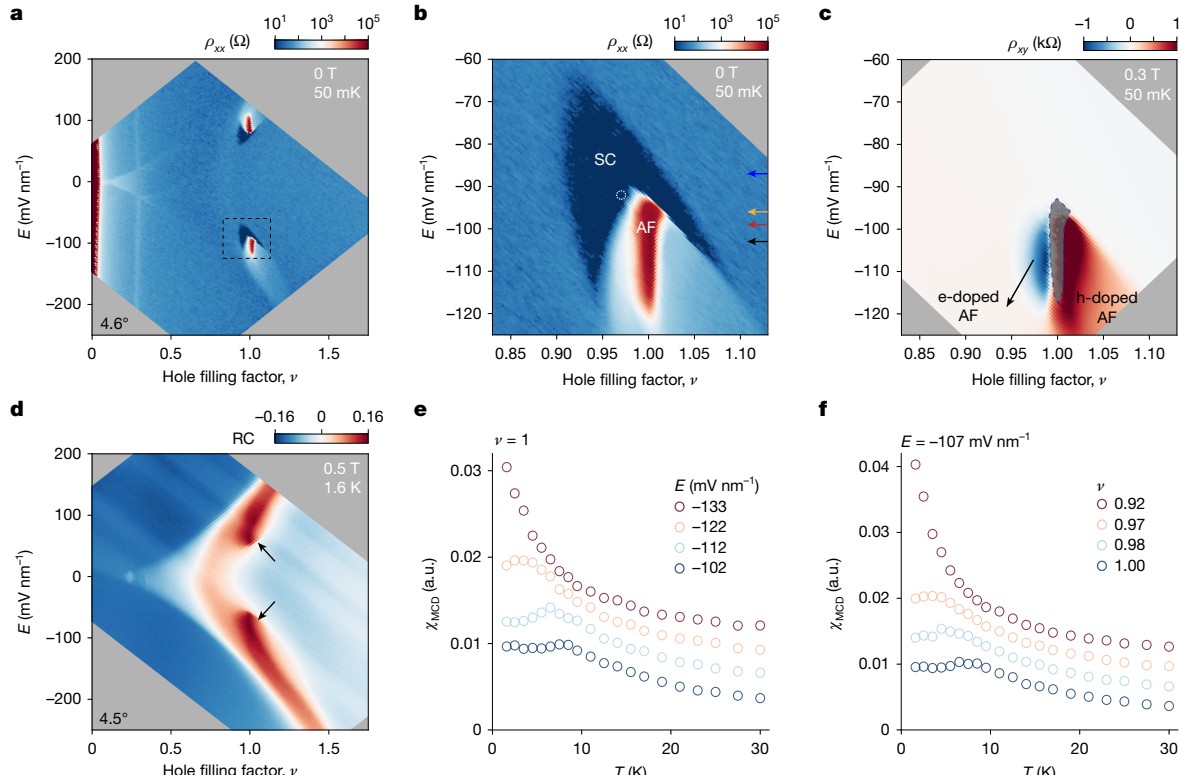

**Fig. 2 | Superconductor and AF insulator. a**, Resistivity as a function of $\nu$ and $E$ at $T = 50$ mK and $B = 0$ T (device 1, 4.6° tWSe$_2$). **b,c**, Close-up view of the dashed box in **a** for resistivity $\rho_{xx}$ (**b**) and Hall resistivity $\rho_{xy}$ at $B = 0.3$ T (**c**). The white circle in **b** marks the location with the highest $T_c$. Arrows denote the $E$-fields applied for the phase diagrams mapped in Fig. 3. **d**, Reflection contrast (RC) at the moiré exciton resonance as a function of $\nu$ and $E$ at $T = 1.6$ K and $B = 0.5$ T

(device 2, 4.5° tWSe$_2$). Arrows denote the AF insulator with enhanced RC. **e,f**, Temperature dependence of magnetic susceptibility at $\nu = 1$ and varying $E$-fields (**e**) and at $E = -107$ mV nm$^{-1}$ and varying filling factors (**f**). The curves are vertically displaced by 0.003 for clarity. In **a**–**d**, the grey regions are experimentally inaccessible.

appears right next to the 'melting point' of the $\nu = 1$ insulator on the low $E$-field end in the layer-hybridized region. The optimal $T_c$ increases with $\theta$ (Fig. 1f,g). At $\nu = 1$, superconductors occupy a narrow strip (dark blue) in the ($\theta$–$E$) phase diagram (Fig. 1e). Both $\theta$ and $E$ can induce an insulator-to-superconductor transition. Theoretical studies examined the phase diagram in the small and large twist angle limits (large and small $U/W$)[30–45]. The moderate $U/W$ regime provides the possibility of studying the intricate phase diagram as in high-$T_c$ materials[2–5].

## Superconductor and AF insulator

We focus on tWSe$_2$ with moderate $U/W$. Figure 2a shows resistivity $\rho_{xx}$ as a function of $E$ and $\nu$ for device 1 ($\theta = 4.6°$) at 50 mK. The $\nu = 1$ insulator is present only near the region intersected by the vHS. Figure 2b is a close-up view of the dashed box in Fig. 2a. Figure 2c shows Hall resistivity $\rho_{xy}$ of the same region measured under an out-of-plane magnetic field of $B = 0.3$ T. The measurement of the Hall resistivity of the insulator (shaded in grey) is not reliable because of its large resistivity. The superconductor wraps around the insulator with an intermediate metallic phase. The metal exhibits a substantially enhanced Hall resistivity, corresponding to small density electron conduction for $\nu < 1$ and hole conduction for $\nu > 1$.

The superconductor is most robust near its boundary with the insulator (Extended Data Fig. 3). The highest $T_c$, the location of which is denoted by an empty circle in Fig. 2b, is about 400 mK. Detailed characterizations of the state are shown in Extended Data Fig. 4. At 50 mK, the critical $B$-field is about 0.15 T. The coherence length is $\xi \approx 34$ nm from the Ginzburg–Landau analysis of the $T$-dependence of the critical field. Given the moiré period is about 4.1 nm and the

electron mean free path ≥400 nm (estimated from the normal-state resistivity), the Cooper pairs are tightly bound, and the superconductor is in the clean limit.

We examine the nature of the insulator at $\nu = 1$ by probing its magnetic susceptibility $\chi_{MCD}$, which was determined as the weak-field slope of the magnetic circular dichroism signal (Methods). Figure 2d shows the reflection contrast of the tWSe$_2$ moiré exciton as a function of $E$ and $\nu$ at 1.6 K for device 2 ($\theta = 4.5°$), which is free of contact and split metal gates for optical access. The insulator (between approximately $-70$ mV nm$^{-1}$ and $-120$ mV nm$^{-1}$) is identified by an enhanced reflection contrast. Figure 2e shows the $T$-dependence of $\chi_{MCD}$ at $\nu = 1$ for selected $E$-fields. The curves are vertically displaced for clarity. The susceptibility for the insulator at $E = -102$ mV nm$^{-1}$ shows a characteristic temperature dependence for an antiferromagnet with a cusp around 8 K. The cusp marks the Néel temperature $T_N$, below which the antiferromagnet orders. For $T > T_N$, the susceptibility follows the Curie–Weiss law with a negative Curie–Weiss temperature, the magnitude of which is also of the order of $T_N$. As $E$ increases, $T_N$ continuously decreases. The susceptibility for the metallic state ($E = -133$ mV nm$^{-1}$) diverges at low temperature, and long-range order is absent. A similar trend is observed in Fig. 2f for doping away from the $\nu = 1$ insulator at a fixed $E$-field (see Extended Data Fig. 5 for more susceptibility data and analyses). These results indicate that the insulator at $\nu = 1$ is an AF insulator and the AF order persists, albeit weakened, when small doping is introduced. Although the exact value of $T_N$ and the $E$-field range for the insulator depend on twist angle and other details (such as unintentional strain in the sample), the AF insulating phase at $\nu = 1$ is generic for tWSe$_2$ with moderate correlation. Theoretical studies showed that this state could be described by

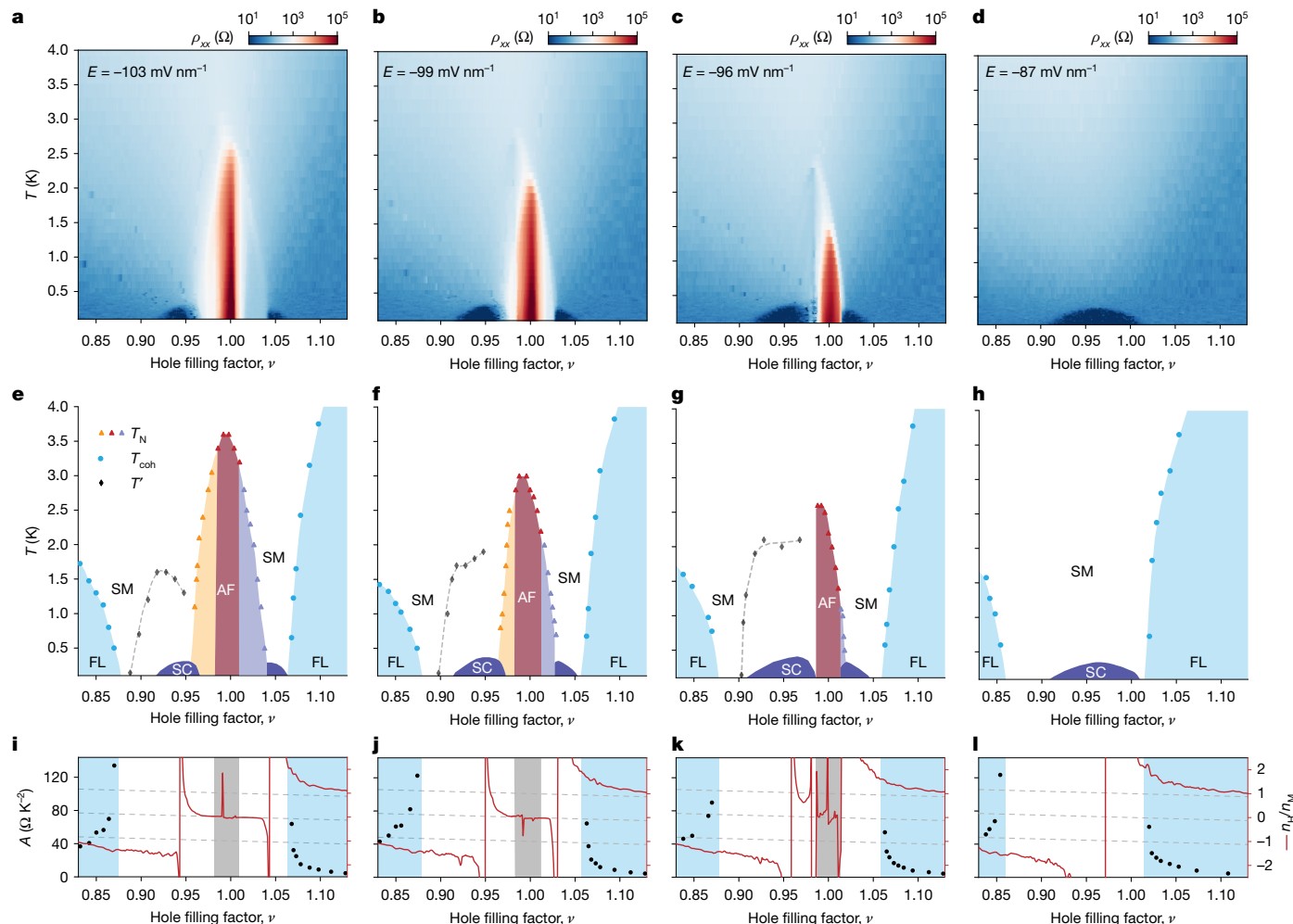

**Fig. 3 | Phase diagrams. a–d**, Resistivity under $B = 0$ T as a function of $\nu$ and $T$. **e–h**, Filling–temperature phase diagram exhibits an AF insulator (brown), an electron-doped AF insulator (yellow), a hole-doped AF insulator (purple), a superconductor (SC) (dark blue), a strange metal (SM) and a Fermi liquid (FL) (light blue) with corresponding temperature scales $T_N$, $T_{coh}$ and $T'$ (symbols). **i–l**, Filling factor dependence of $A$ (left axis, symbols, from the $T$–square

resistivity analysis) and Hall density $n_H/n_M$ (right axis, red lines, from Hall resistivity under $B = 0.3$ T at $T = 50$ mK). Hall density cannot be determined reliably in the grey-shaded region for the AF insulator. The dashed lines denote Hall density $2 - \nu$, $1 - \nu$ and $-\nu$, each separated by one moiré density. The panels from left to right correspond to $E = -103$ mV nm$^{-1}$, $-99$ mV nm$^{-1}$, $-96$ mV nm$^{-1}$ and $-87$ mV nm$^{-1}$. The critical field for the Mott transition is about $-92$ mV nm$^{-1}$.

the 2D anisotropic XXZ model with an $E$-dependent Dzyaloshinskii–Moriya interaction term, and the spin anisotropy favours a 120° Néel order[9–11,29].

## Phase diagram

We map the phase diagram of tWSe$_2$ (device 1) at four representative $E$-fields across the Mott transition as denoted by the arrows in Fig. 2b. Figure 3 shows resistivity at $B = 0$ T as a function of temperature and filling (Fig. 3a–d), the temperature–filling phase diagrams (Fig. 3e–h) and the Hall density versus filling at 50 mK (Fig. 3i–l). The temperature–$E$-field phase diagram at $\nu = 1$ is included in Extended Data Fig. 10. We first consider the case of $E = -103$ mV nm$^{-1}$. The resistivity map (Fig. 3a) shows a prominent insulator at $\nu = 1$ and a superconducting dome on each side of the insulator. The superconductor below $\nu = 1$ is more robust. The regions between the insulator and the superconductors show enhanced Hall resistivity at low temperature (Extended Data Fig. 6). Detailed analysis of the temperature dependence of resistivity is shown in Fig. 4 and Extended Data Fig. 6 (see Extended Data Figs. 7–9 for the other $E$-fields).

At $\nu = 1$, $\rho_{xx}$ decreases with decreasing temperature till about 3.5 K, below which it increases sharply by nearly three orders of magnitude

(Fig. 4a). Together with the magnetic susceptibility result above, this supports an AF insulator at low temperature. We estimated $T_N \approx 3.5$ K from the local resistivity minimum. When small doping is introduced, the sample becomes metallic (Fig. 4b,c): $\rho_{xx}$ decreases with decreasing temperature except for a sharp peak or rise around 2.5 K, which is accompanied by an abrupt jump in the Hall density $n_H$. At low temperature, the Hall density, in units of the moiré density $n_M$, approaches 0.02 and −0.02 (denoted by dashed lines) for $\nu = 0.98$ and 1.02, respectively. This supports a phase transition between two distinct metallic states with an abrupt Fermi surface reconstruction. The low-temperature phase at filling $\nu$ has a small Fermi surface with carrier density $(1 - \nu)$ and is antiferromagnetically ordered (Fig. 2f). It is a doped AF insulator; the Mott gap survives on doping[3]. We estimated $T_N$ from the onset of the Hall density jump or local resistivity minimum. The Néel temperature continuously decreases with doping away from the AF insulator and vanishes near $\nu = 0.96$ and 1.04 (Fig. 3e).

On the electron doping side, for a wide filling range $0.88 \leq \nu \leq 0.96$, covering the entire superconducting dome, $\rho_{xx}$ follows a $T$-linear dependence above temperature $T'$ and a sublinear dependence below $T'$ (Fig. 4d). We define the crossover temperature scale $T'$ as the temperature at which $\rho_{xx}$ deviates from the $T$-linear dependence by 10%. $T'$ has

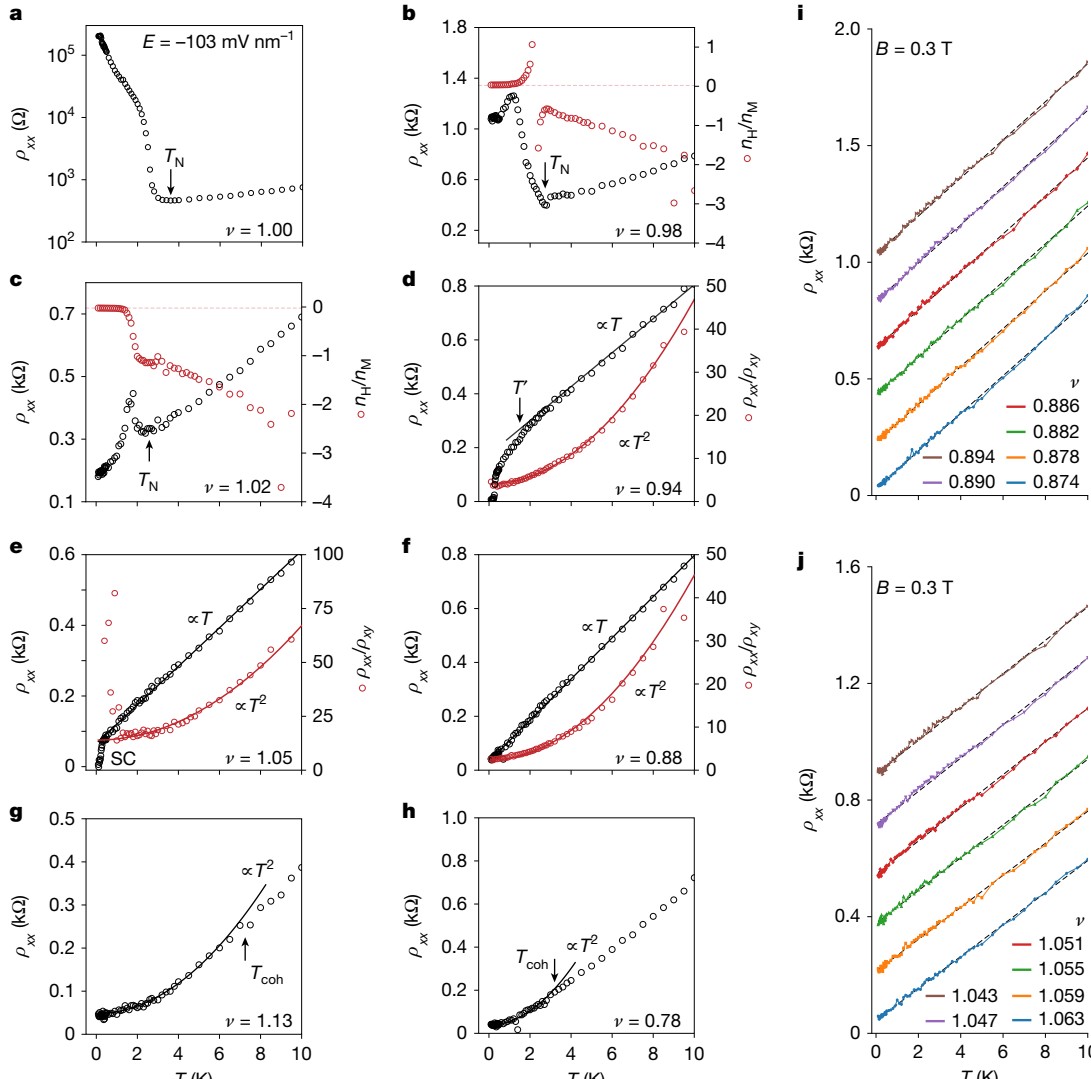

**Fig. 4 | Temperature-dependent transport at $E = -103$ mV nm$^{-1}$.**
**a**–**h**, Temperature dependence of the transport characteristics (device 1, 4.6° tWSe$_2$) at $v = 1$ (**a**), 0.98 (**b**), 1.02 (**c**), 0.94 (**d**), 1.05 (**e**), 0.88 (**f**), 1.13 (**g**) and 0.78 (**h**). Left axis denotes resistivity $\rho_{xx}$ under $B = 0$ T (black symbols for experiment and black lines for linear or quadratic fits); right axis denotes $n_H/n_M$ (**b**,**c**) and $\rho_{xx}/\rho_{xy}$ (**d**–**f**) determined under $B = 0.3$ T (red symbols for experiment and red solid lines for quadratic fits). Red dashed lines in **b** and **c** denote Hall density $(1 - v)$. The Néel temperature $T_N$ was estimated from the local resistivity

minimum. The crossover temperature $T'$ was estimated at 10% deviation from the $T$–linear resistivity. The coherence temperature $T_{coh}$ was estimated at 10% deviation from the $T$–square resistivity. **i**,**j**, $T$–linear resistivity at different filling factors for $v < 1$ (**i**) and $v > 1$ (**j**). A small magnetic field of $B = 0.3$ T was applied to quench the superconductivity. The curves are vertically displaced by 0.2 kΩ (**i**) and 0.16 kΩ (**j**) for clarity. Black dashed lines are linear fits to the data (symbols).

a dome-like dependence on filling and vanishes near $v = 0.88$ (Fig. 3e). Below $v = 0.88$, a Fermi liquid behaviour emerges with resistivity following $\rho_{xx} = \rho_0 + AT^2$ (Fig. 4h), where $\rho_0$ is the residual resistivity and coefficient $A^{1/2}$ is often used to track the quasiparticle effective mass. The resistivity deviates from the $T^2$-dependence above the coherence temperature $T_{coh}$, which was estimated at 10% deviation. As $v$ approaches 0.88 from below, $T_{coh}$ decreases and vanishes (Fig. 3e). Accordingly, $A$ increases substantially (black symbols, Fig. 3i). A filling window of about 0.02 near $v = 0.88$ separates the Fermi liquid and the $T$-sublinear metal. Here we observe $T$–linear resistivity over a wide temperature range (100 mK to 10 K) (Fig. 4i), as well as $T$-square $\rho_{xx}/\rho_{xy}$, which relates to the inverse Hall angle (Fig. 4f). These behaviours are incompatible with the quasiparticle description of metallic conduction[46]; they are rather like those of 'strange metals' found in strongly correlated materials[12–14,46]. The $T$–linear resistivity shows a slope around 100 Ω K$^{-1}$ and a residual resistivity below 100 Ω, belonging to the clean limit of strange metals (see Methods for further discussions).

On the hole doping side ($v > 1$), the phase diagram is similar except that the temperature scale $T'$ and the $T$-sublinear metal can no longer be identified (Fig. 4c,e and Extended Data Fig. 6). $T$–linear resistivity and $T$–square $\rho_{xx}/\rho_{xy}$ are also observed over an extended filling range, covering the entire superconducting dome; a small magnetic field of $B = 0.3$ T was applied to quench superconductivity to enable measurements down to the lowest temperature in this study (Fig. 4e,j).

The phase diagram (Fig. 3e) bears a resemblance to that of the high-$T_c$ cuprates[2–5]. Doping an AF insulator with a small number of electrons or holes yields an AF metal with a small Fermi surface. Near the optimal doping for superconductivity, the Mott gap collapses, manifesting a Hall density jump[47] by about $n_M$ (Fig. 3i). Right before the collapse of the Mott gap, $T_c$ is about 3.5% of the effective Fermi temperature (Methods), on par with most unconventional superconductors[48]. Subtle differences from the cuprate phase diagram are also noted. For instance, the strange metal phase near $v = 0.88$ is outside the

superconducting dome. These differences may arise from the different lattice symmetry (triangular compared with square) and require further studies.

## Evolution through the Mott transition

Last, we examine the evolution of the phase diagram through the $E$-field-tuned Mott transition (critical field $E_c \approx -92$ mV nm$^{-1}$). As $E$-field varies from $-103$ mV nm$^{-1}$ (Fig. 3a,e,i) to $-99$ mV nm$^{-1}$ (Fig. 3b,f,j) to $-96$ mV nm$^{-1}$ (Fig. 3c,g,k), the system moves closer to the Mott transition from the insulator side. The phase diagram remains qualitatively unchanged but $T_N$ decreases; the doped AF insulator shrinks; the superconducting domes expand and $T_c$ increases. Also observed are the Hall density jumps by about $n_M$ near the optimal doping levels and the Fermi liquids at sufficiently large doping levels (the onset of which is marked by a substantial increase in $A$). However, across the Mott transition on the metal side ($E = -87$ mV nm$^{-1}$), the AF insulator disappears and only one superconducting dome remains (Fig. 3d,h,l). The Hall density jump at optimal doping is about $2\,n_M$, which is associated with the vHS in the band structure. Compared with the strange metal behaviour in Fig. 4i,j, the $T$–linear resistivity here is observed over an even wider filling range $0.86 \leq \nu \leq 1.01$. The result indicates that strange metallicity could occur over a range of tuning parameters instead of a quantum critical point[49] (see Extended Data Figs. 9 and 10 for further results and analysis). This highlights the complexity of the phase diagram near the Mott transition and calls for experiments beyond d.c. transport. We summarize the interplay between the AF order and superconductivity in Extended Data Fig. 11. As $E$ approaches $E_c$ from the insulator side, $T_N$ decreases, and optimal $T_c$ increases monotonically; $T_c/T_N$ increases from 4% (at $-110$ mV nm$^{-1}$) to 18% (at $-96$ mV nm$^{-1}$) and diverges at $E_c$.

## Conclusion and outlook

Tuning tWSe$_2$ to the moderate correlation regime through its twist angle, we observe a phase diagram around $\nu = 1$ that resembles the iconic phase diagram of the high-$T_c$ cuprates[2–5]. Superconductivity is observed only near the Mott transition tuned by both doping and bandwidth, which is consistent with numerical studies of the Hubbard model[50]. The observations suggest that pure phonon-mediated Cooper pairing is unlikely and promote the importance of strong correlation for high-$T_c$ superconductivity. More experimental probes are required to further understand the complex phase diagram. The platform based on TMD moiré superlattices may provide new perspectives on the high-$T_c$ and strange metal problems.

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

## Methods

### Device geometry and fabrication

The tWSe$_2$ field-effect devices use contact gates and split gates (Fig. 1a and Extended Data Fig. 1) to achieve low contact resistances (typically about 4 kΩ) and to turn off unwanted parallel conduction channels, respectively. The device channel is defined by gates, including the top, bottom, contact and split gates. Details of the device geometry and device fabrication have been described in ref. 7. In brief, 2D flakes, including few-layer graphite, hBN and monolayer WSe$_2$, were mechanically exfoliated from bulk crystals onto silicon substrates. Large WSe$_2$ monolayers were cut into half by an atomic force microscope tip. These flakes were sequentially picked up using a polycarbonate stamp. From top to bottom, the heterostructures consist of a hBN capping layer, a graphite and hBN (3.4 nm) top gate, the first and second half of a WSe$_2$ monolayer with a small twist angle, and a hBN (12.0 nm) and graphite bottom gate. The heterostructures were released onto silicon substrates with pre-patterned titanium/platinum (5 nm/25 nm) electrodes. Titanium/palladium (5 nm/35 nm) contact gates and split gates were deposited using the standard electron-beam lithography and evaporation techniques.

### Electrical measurements

The electrical transport measurements were performed in a dilution refrigerator (Bluefors LD250) equipped with a 12 T superconducting magnet. Silver-epoxy filters (Basel Precision Instruments MFT25) and additional resistor–capacitor filters were installed on the mixing chamber plate to ensure sufficient thermalization and efficient filtering of the high-frequency radiation, achieving a base electron temperature of less than 100 mK. The longitudinal and transverse resistances were measured using the standard low-frequency (5.777 Hz) lock-in technique. The excitation current was kept at 1 nA to minimize sample heating. The sample resistivity, $\rho_{xx} = R_{xx}/(L/W)$, was calculated from the longitudinal resistance $R_{xx}$, where $L$ (≈1.6 μm) is the centre-to-centre distance between the voltage probes and $W$ (≈1.3 μm) is the channel width. The Hall resistivity $\rho_{xy}$ is the anti-symmetrized transverse resistance $R_{xy}$ under a positive and negative $B$-field.

### Magneto-optical measurements

Magneto-optical measurements were performed in a closed-cycle cryostat (attocube, attoDRY 2100) down to 1.6 K. Details of the measurements have been reported in ref. 22. In short, a linear polarizer and a quarter-wave plate were used to generate circularly polarized light ($\sigma^+$ and $\sigma^-$) from a light-emitting diode. The polarized light beam was focused onto samples by a low-temperature objective lens (numerical aperture 0.8) with a spot size of about 1 μm. The excitation intensity was kept below 50 nW μm$^{-2}$ to minimize sample heating (no measurable changes in the MCD were observed on further reduction of the excitation power by an order of magnitude). The reflected light was collected by the same objective and directed to a spectrometer equipped with a liquid-nitrogen-cooled charge-coupled device for spectral acquisition. The MCD spectrum is defined as $(I^- - I^+)/(I^- + I^+)$, where $I^-$ and $I^+$ are the reflection spectra for the $\sigma^-$ and $\sigma^+$ incident light, respectively. To obtain the data shown in the figures, we integrated the MCD signal over a narrow spectral window (735–740 nm) covering the moiré exciton resonance of tWSe$_2$. The magnetic susceptibility was extracted from the slope of the $B$-field dependence of the MCD signal at zero field. The reflection contrast spectrum is defined as $(I - I_0)/I_0$, where $I$ is the raw reflection spectrum and $I_0$ is the reference spectrum at a high doping density with no distinct excitonic resonances. To obtain the reflection contrast map in Fig. 2d, we used the mean value of the reflection contrast over the spectral window 736–741 nm that covers the moiré exciton resonance.

### Band structure calculations

The low-energy electronic band structure of small twist angle tWSe$_2$ was calculated using the continuum model[26,27]. Monolayer WSe$_2$ is a direct-gap semiconductor with the bandgap located at the two inequivalent corners of the hexagonal BZ, K and K′. The electron valley degree of freedom is locked to the spin degree of freedom because of the broken inversion symmetry and large spin–orbit coupling[51]. The K and K′ states, carrying opposite spin polarizations, are related by time-reversal symmetry. In tWSe$_2$, the valley pocket K$_t$ and K$_b$, originated from the top and bottom monolayers, respectively, are slightly displaced in momentum; they define the corners of the moiré BZ for each spin flavour. The low-energy physics of hole-doped tWSe$_2$ is captured by a two-band $\mathbf{k} \cdot \mathbf{p}$ model within an effective mass description.

The effective moiré Hamiltonian for the K-valley (spin-up) states can be written as

$$H_\uparrow = \begin{pmatrix} -\dfrac{\hbar^2(\mathbf{k} - \boldsymbol{\kappa}_+)^2}{2m^*} + \Delta_t(\mathbf{r}) & \Delta_T(\mathbf{r}) \\[2mm] \Delta_T^\dagger(\mathbf{r}) & -\dfrac{\hbar^2(\mathbf{k} - \boldsymbol{\kappa}_-)^2}{2m^*} + \Delta_b(\mathbf{r}) \end{pmatrix}. \tag{1}$$

The Hamiltonian $H_\downarrow$ for the K′-valley (spin-down) states can be obtained by a time-reversal transformation of $H_\uparrow$. In equation (1), $\mathbf{k}$ and $\mathbf{r}$ denote, respectively, the wave vector and position vector of the holes; $\boldsymbol{\kappa}_\pm$ represent the corners of the moiré BZ; $\hbar$ is the reduced Planck constant; $m^* \approx 0.45m_0$ is the hole effective mass for the highest valence band of monolayer WSe$_2$ with $m_0$ denoting the free electron mass[52]. The holes are subjected to a periodic pseudomagnetic field, $\Delta(\mathbf{r}) = (\mathrm{Re}\Delta_T^\dagger, \ \mathrm{Im}\Delta_T^\dagger, \ \frac{\Delta_t - \Delta_b}{2})$, with the moiré period. Within the lowest harmonic approximation, $\Delta_T(\mathbf{r}) = w(1 + e^{-i\mathbf{g}_2 \cdot \mathbf{r}} + e^{-i\mathbf{g}_3 \cdot \mathbf{r}})$ is the interlayer tunnelling amplitude, and $\Delta_{t,b}(\mathbf{r}) = \pm\frac{V_z}{2} + 2V\sum_{j=1,3,5}\cos(\mathbf{g}_j \cdot \mathbf{r} \pm \psi)$ are the intralayer moiré potentials for the top and bottom layers, respectively. Here $V_z$ is the sublattice potential difference; $\mathbf{g}_{j=1,2,3}$ are the reciprocal lattice vectors obtained by rotating $\mathbf{g}_1 = (\frac{4\pi}{\sqrt{3}a_M}, 0)$ anticlockwise by an angle $(j-1)\pi/3$. The parameters $(V, \psi, w) = $ (13.6 meV, 49.1°, 10.0 meV) are adopted from a scanning tunnelling microscopy study on tWSe$_2$ (ref. 28).

The band structure was obtained by diagonalizing the Hamiltonian, and the DOS was computed from the band structure. For comparison with experiment, the sublattice potential difference was converted to $E$-field using the relation, $E = V_z/(\frac{\varepsilon_{\mathrm{hBN}}}{\varepsilon_{\mathrm{TMD}}}ed)$, where the dipole moment $\frac{\varepsilon_{\mathrm{hBN}}}{\varepsilon_{\mathrm{TMD}}}ed \approx 0.26e$ nm was independently determined from the anticrossing feature of the layer-hybridized moiré excitons, following refs. 53,54. The value is consistent with the accepted out-of-plane dielectric constants of hBN and WSe$_2$ ($\varepsilon_{\mathrm{hBN}} \approx 3$ and $\varepsilon_{\mathrm{TMD}} \approx 8$) and the interlayer separation in WSe$_2$ ($d \approx 0.7$ nm).

### $E$-field dependence of the $v = 1$ insulator

We provide a qualitative understanding of the melting behaviour of the $v = 1$ insulator as twist angle increases (Fig. 1d). The $E$-field tunes the location of the vHS in the $E$–$v$ phase diagram (Fig. 1b,c). For fixed filling at $v = 1$, the $E$-field tunes the vHS relative to the Fermi level. The closer the vHS is to the Fermi level, the lower the Fermi velocity gets, and the more stable the $v = 1$ insulator is. In samples with intermediate $\theta \approx 3.6$–$4.6°$, the $v = 1$ insulator is absent at both small and large $E$-fields; it is stabilized only at intermediate $E$-fields, in which the vHS is sufficiently close to the Fermi level. For $\theta \geq 4.7°$, the correlation is too weak to stabilize the insulator even with the help of the vHS. For $\theta \leq 3.5°$, the strong correlation can stabilize the insulator in the entire layer-hybridized region regardless of the location of the vHS. Effectively, both $\theta$ and $E$ can tune the band flatness at the Fermi level and realize a bandwidth-tuned Mott transition at $v = 1$.

### High-$T_c$ superconductivity

The observed superconductivity in the under-doped regime is of high relative $T_c$. First, we estimate $T_c/T_F$ at $v \approx 1.023$ and $E = -99$ mV nm$^{-1}$, at which the superconducting dome intersects the AF metal, and $T_c \approx 0.3$ K

is the highest on the hole doping side (Extended Data Fig. 12). We estimate $T_F$ from the charge density and effective mass using the parabolic band approximation. In the under-doped regime, superconductivity emerges from the small Fermi surface AF metal and the density was determined to be $(\nu - 1)n_M \approx 0.023\,n_M$ ($n_M \approx 6.7 \times 10^{12}\,\mathrm{cm}^{-2}$). The effective mass can be strongly renormalized by interactions. We determined it from the $T$-dependent Shubnikov–de Haas oscillations (Extended Data Fig. 2). At $\nu \approx 1.023$, under relatively low magnetic fields, clear quantum oscillations start to emerge only when the $E$-field is tuned slightly away from $E = -99\,\mathrm{mV\,nm}^{-1}$, indicating that the effective mass decreases. We used the effective mass $m^* \approx 0.5m_0$ at $E = -105\,\mathrm{mV\,nm}^{-1}$ as an estimate of its lower bound. This leads to $T_F \approx \frac{\hbar^2\pi^* \cdot 0.023 n_M}{m^* k_B} \approx 8.5\,\mathrm{K}$ and $T_c/T_F \approx 3.5\%$ (lower bound), which is on par with most unconventional superconductors[48] ($k_B$ is the Boltzmann constant). Second, the ratio of the coherence length ($\xi \approx 34\,\mathrm{nm}$) to the lattice period is about 8.3 (if the moiré period is used) or 1.4 (if the inter-particle spacing $\frac{1}{\sqrt{n_H}}$ is used). Both values are comparable to 5, the typical ratio for cuprates[55], supporting tightly bound Cooper pairs in tWSe$_2$. Last, the optimal $T_c/T_N$ determined at each $E$-field varies from 4% to 18% (Extended Data Fig. 11). This ratio, which is also comparable to that in cuprates, further supports high-$T_c$ superconductivity in tWSe$_2$.

## Hall density jumps

Abrupt Hall density jumps are observed in Fig. 3i–l. The three dashed lines illustrate filling dependences of the Hall density $n_H/n_M = 2 - \nu$, $1 - \nu$ and $-\nu$. On the insulator side of the Mott transition at $\nu = 1$ (Fig. 3i–k), the Hall density follows $(1 - \nu)$ in the vicinity of $\nu = 1$. It jumps at the peak of the superconducting domes, and away from the jumps, follows $(2 - \nu)$ for $\nu > 1$ and $-\nu$ for $\nu < 1$. The step of each jump is $n_M$. We highlight that the observed Hall density jumps by $n_M$ are tied to the peak of the superconducting domes; they are distinct from the reported Hall density jumps in graphene moiré systems, which typically occur at integer fillings[17,56,57]. On the metal side of the Mott transition at $\nu = 1$ (Fig. 3l), the small Fermi surface state is absent. There is only one jump in the Hall density that is tied to the vHS. The Hall density follows $(2 - \nu)$ and $-\nu$ on two sides of the jump; the corresponding step is $2\,n_M$.

## Estimate of the Bloch–Grüneisen temperature

The observed $T$–linear resistivity over a wide temperature range from 100 mK to 10 K (Fig. 4 and Extended Data Figs. 6–10) cannot be explained by electron–phonon scattering because this behaviour is expected only above the Bloch–Grüneisen temperature $T_{BG} = \frac{2\hbar v_p k_F}{k_B}$. For tWSe$_2$, $T_{BG}$ was estimated to be 17.5 K $\gg$ 100 mK (ref. 58), where phonon velocity $v_p \approx 2{,}510\,\mathrm{m\,s}^{-1}$ and Fermi wavevector $k_F \approx \sqrt{\pi n_M}$ were used near $\nu = 1$. Next, the observed $T$-linear $\rho_{xx}$ and $T$-square $\rho_{xx}/\rho_{xy}$ (Fig. 4) are not compatible with the usual quasiparticle picture, which would predict $\rho_{xx} \propto \tau^{-1} \propto T$, $\rho_{xy}$ independent of $t$ and $T$, and therefore $\rho_{xx}/\rho_{xy} \propto \tau^{-1} \propto T$ (ref. 46). Last, $T$–linear resistivity is observed only near the superconducting domes, which is not compatible with the

electron–phonon mechanism. We also note that the temperature scale $T'$ is distinct from $T_{BG}$ because $T'$ ($\leq$2 K) is substantially smaller than $T_{BG}$, strongly dependent on doping over a narrow doping range, and below it, resistivity is sublinear in $T$. By contrast, $T_{BG}$ is nearly a constant for the small doping range, and below it, resistivity is super-linear in $T$.

## Data availability

Data are available from the corresponding authors upon request. Source data are provided with this paper.

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

**Acknowledgements** We thank P. A. Lee, A. J. Millis, L. Fu, P. Coleman, S. Todadri, Q. Si, D. Chowdhury, A. Vishwanath, B. A. Bernevig, J. H. Pixley, A. H. MacDonald and P. W. Phillips for discussions. This work was supported by the Gordon and Betty Moore Foundation (grant https://doi.org/10.37807/GBMF11563, transport measurements) and the Air Force Office of Scientific Research under award no. FA9550-20-1-0219 (magneto-optical measurements). We used the Cornell NanoScale Facility, an NNCI member supported by NSF grant NNCI-2025233, for sample fabrication. A portion of this work was performed at the National High Magnetic Field Laboratory, which is supported by the National Science Foundation cooperative agreement no. DMR-2128556* and the State of Florida. The growth of the hBN crystals was supported by the Elemental Strategy Initiative of MEXT, Japan, and CREST (JPMJCR15F3), JST. We also acknowledge support from the David and Lucile Packard Fellowship (K.F.M.).

**Author contributions** Y.X., Z.H., J.Z. and Y.Z. fabricated the devices. Y.X., Z.H. and Y.Z. performed the electrical transport measurements and analysed the data. J.Z. and P.K. performed the optical measurements. Z.H. performed the band structure calculations. K.W. and T.T. grew the bulk hBN crystals. K.F.M. and J.S. designed the scientific objectives and oversaw the project. All authors discussed the results and commented on the paper.

**Funding** Open access funding provided by Max Planck Society.

**Competing interests** The authors declare no competing interests.

**Additional information**
**Correspondence and requests for materials** should be addressed to Yiyu Xia, Zhongdong Han, Kin Fai Mak or Jie Shan.

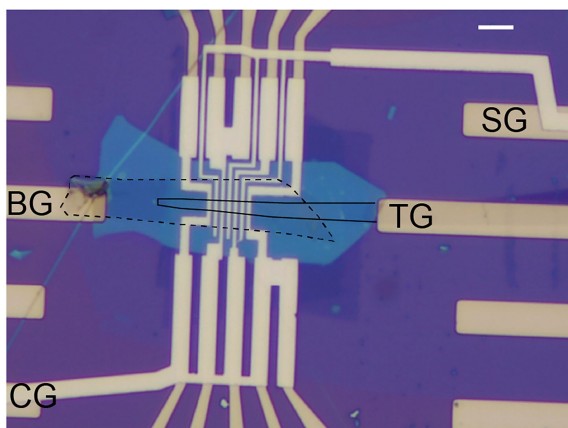

**Extended Data Fig. 1 | Device image.** Optical micrograph image of device 1 (4.6° tWSe$_2$). The scale bar is 4 µm. The device channel is defined by the overlap area of TG (top gate, solid line) and BG (bottom gate, dashed line). CG: contact gate; SG: split gate.

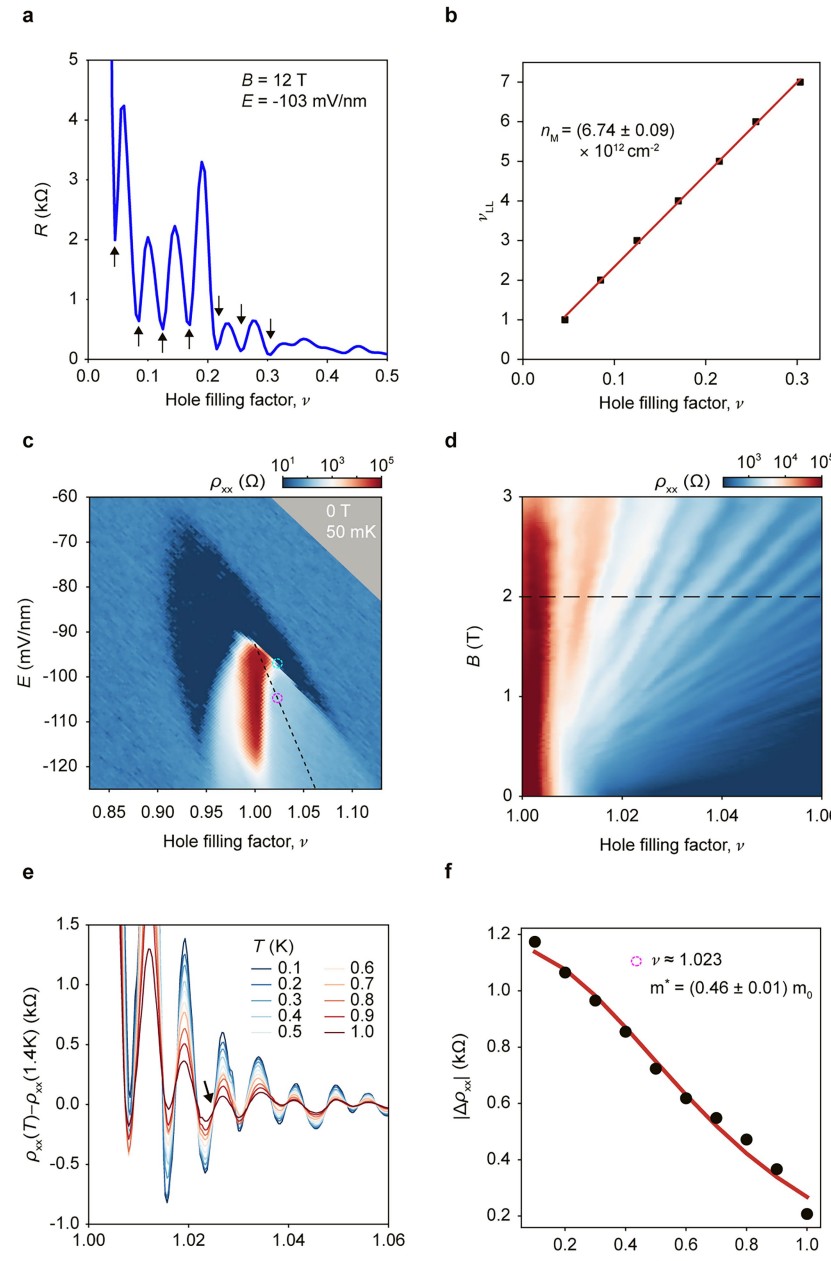

**Extended Data Fig. 2 | Determination of the moiré density and quasiparticle effective mass. a**, Longitudinal resistance $R$ versus hole filling factor $\nu$ at $B = 12$ T, $E = -103$ mV/nm, and $T = 1.6$ K. The arrows denote the Landau level 1–7. **b**, Landau level index $\nu_{LL}$ versus filling follows a linear dependence (red line). The moiré density was determined from the slope. **c**, Resistivity versus $\nu$ and $E$ at $T = 50$ mK and $B = 0$ T. Blue circle: location with coexisting superconductivity and AF; pink circle: location where the quasiparticle effective mass $m^*$ was determined in **f. d**, Resistivity versus $\nu$ and $B$ measured along the black dashed line in **c**. A Landau fan emerges from $\nu = 1$, demonstrating a small Fermi surface with carrier density $(\nu - 1)n_M$. **e**, Linecut of **d** along the dashed line ($B = 2$ T) at varying temperatures. An offset $\rho_{xx}(\nu)$ at $T = 1.4$ K, for which the quantum oscillations are washed out, was introduced to center $\rho_{xx}(\nu)$ around zero. The arrow marks $\nu \approx 1.023$. **f**, Temperature dependence of $|\Delta\rho_{xx}|$, which measures the size of resistivity oscillations (from the dip to the peak around $\nu \approx 1.023$). Red line: fit to data using $|\Delta\rho_{xx}| = \frac{R_a\lambda(T)}{\sinh\lambda(T)}$, where $\lambda(T) = \frac{2\pi^2 k_B T m^*}{\hbar eB}$ with $m^*$ and $R_a$ as free fitting parameters. Data collected from device 1.

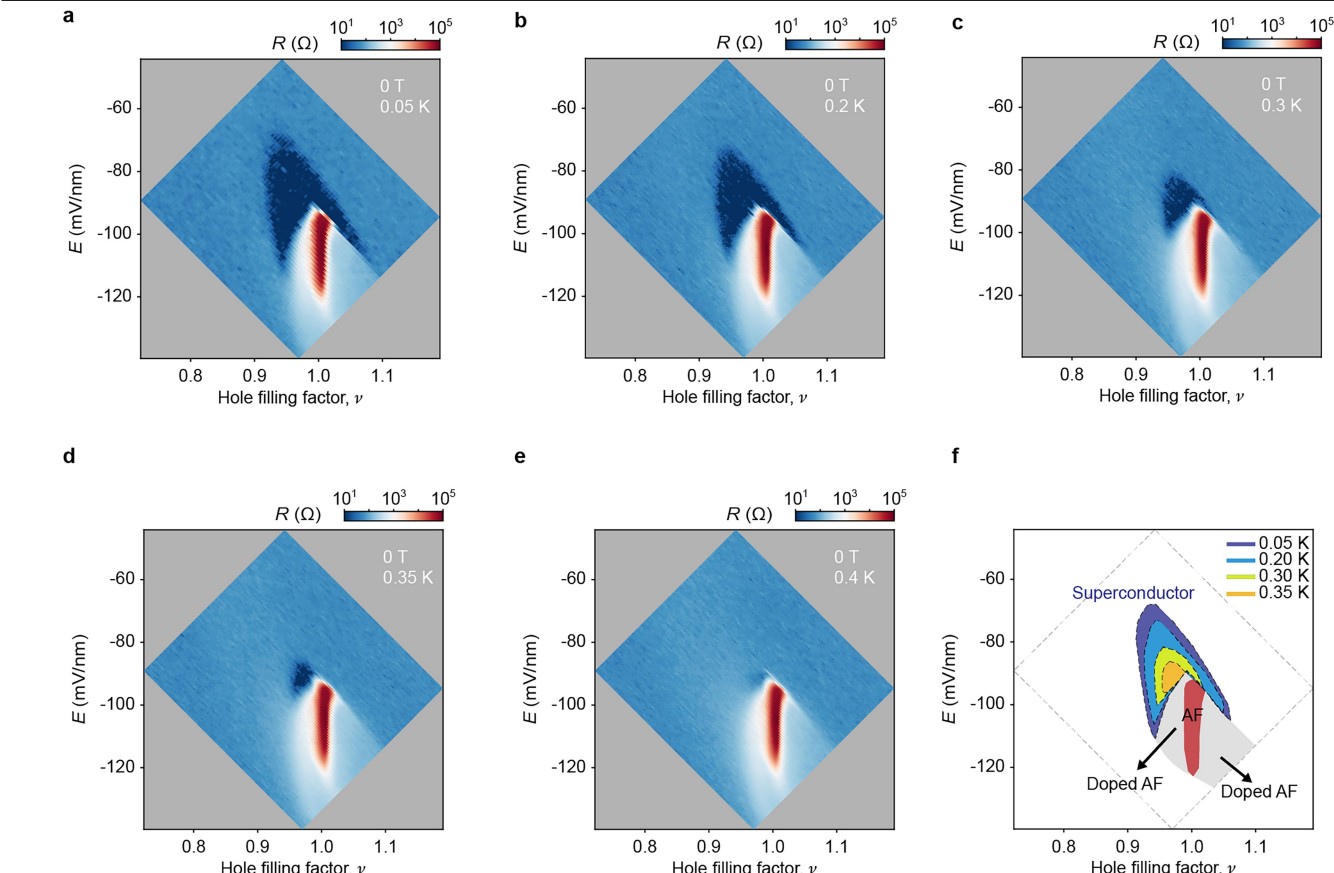

**Extended Data Fig. 3 | Temperature dependent $\nu$ - $E$ phase diagrams. a-e**, Longitudinal resistance $R$ versus $\nu$ and $E$ at $T$ = 0.05 K (**a**), 0.2 K (**b**), 0.3 K (**c**), 0.35 K (**d**), and 0.4 K (**e**) under zero magnetic field. **f**, Coarse map of $T_c$. The region immediately next to the tip of the AF insulator has the highest $T_c$. Data collected from device 1.

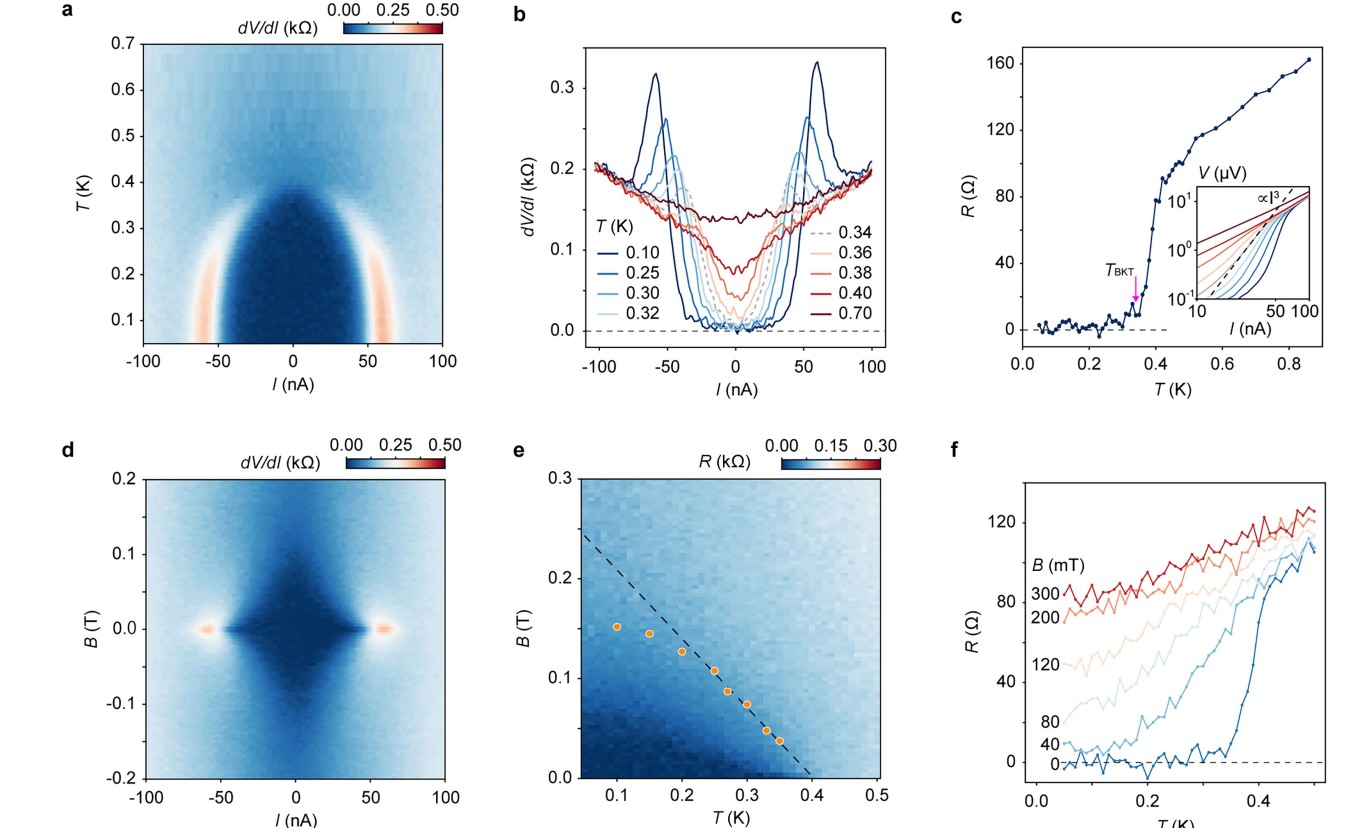

**Extended Data Fig. 4 | Superconductivity with optimal $T_c$ ($v \approx 0.97$, $E \approx -92$ mV/nm). a**, Differential resistance as a function of $T$ and DC bias current $I$ at $B = 0$ T. **b**, Linecuts of **a** at representative temperatures (grey dashed line for the BKT transition temperature $T_{BKT} = 340$ mK). **c**, Temperature dependence of zero-bias resistance $R$. The red arrow denotes $T_{BKT}$. Inset: longitudinal voltage $V$ versus DC bias current $I$ with line color defined in **b**. At $T_{BKT}$, the dependence follows $V \propto I^3$ (dashed line). **d**, Differential resistance versus $B$ and $I$ at 50 mK. **e**, Zero-bias resistance versus $B$ and $T$. Symbols: critical $B$-field $B_{c2}$; dashed line: fit to data using $B_{c2} \approx \frac{\Phi_0}{2\pi\xi^2}\left(1 - \frac{T}{T_c}\right)$, from which the superconducting coherence length $\xi \approx 34$ nm is determined. **f**, Linecuts of **e** at representative magnetic fields. Data collected from device 1.

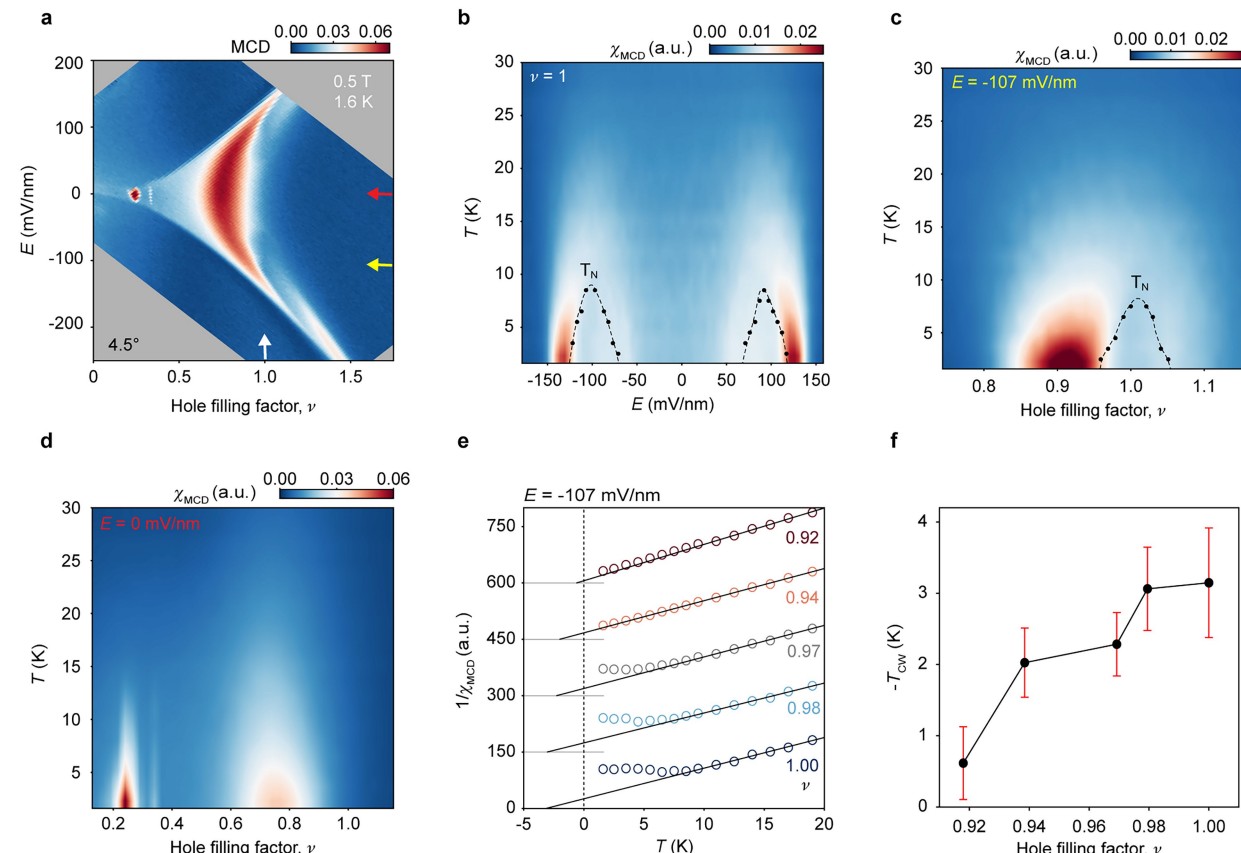

**Extended Data Fig. 5 | Magneto-optical characterizations. a**, MCD versus $\nu$ and $E$ at $T = 1.6$ K and $B = 0.5$ T. MCD is suppressed for the $\nu = 1$ insulator. **b**, Magnetic susceptibility $\chi_{MCD}$ versus $E$ and $T$ at $\nu = 1$. **c,d**, $\chi_{MCD}$ versus $\nu$ and $T$ at $E = -107$ mV/nm (**c**) and $E = 0$ mV/nm (**d**). It is suppressed in regions bound by black dashed lines in **b** and **c**, where AF order emerges. AF ordering is observed in the vicinity of $\nu = 1$ at finite $E$-fields. The enhanced $\chi_{MCD}$ near $\nu = 0.9$ in **c** and near $\nu = 0.8$ in **d** is correlated with the vHS. The generalized Wigner crystals at $\nu = 1/4$ and $1/3$ in **d** also show enhanced $\chi_{MCD}$ because of the local magnetic moments. **e**, $T$-dependence of $1/\chi_{MCD}$ at $E = -107$ mV/nm. The curves for different filling factors are vertically displaced by 150 units for clarity. Solid lines are Curie-Weiss fits to the high-temperature part of the data. **f**, Filling factor dependence of the Curie-Weiss temperature. The negative value reflects the presence of AF interactions, which weaken with doping away from $\nu = 1$. Data collected from device 2.

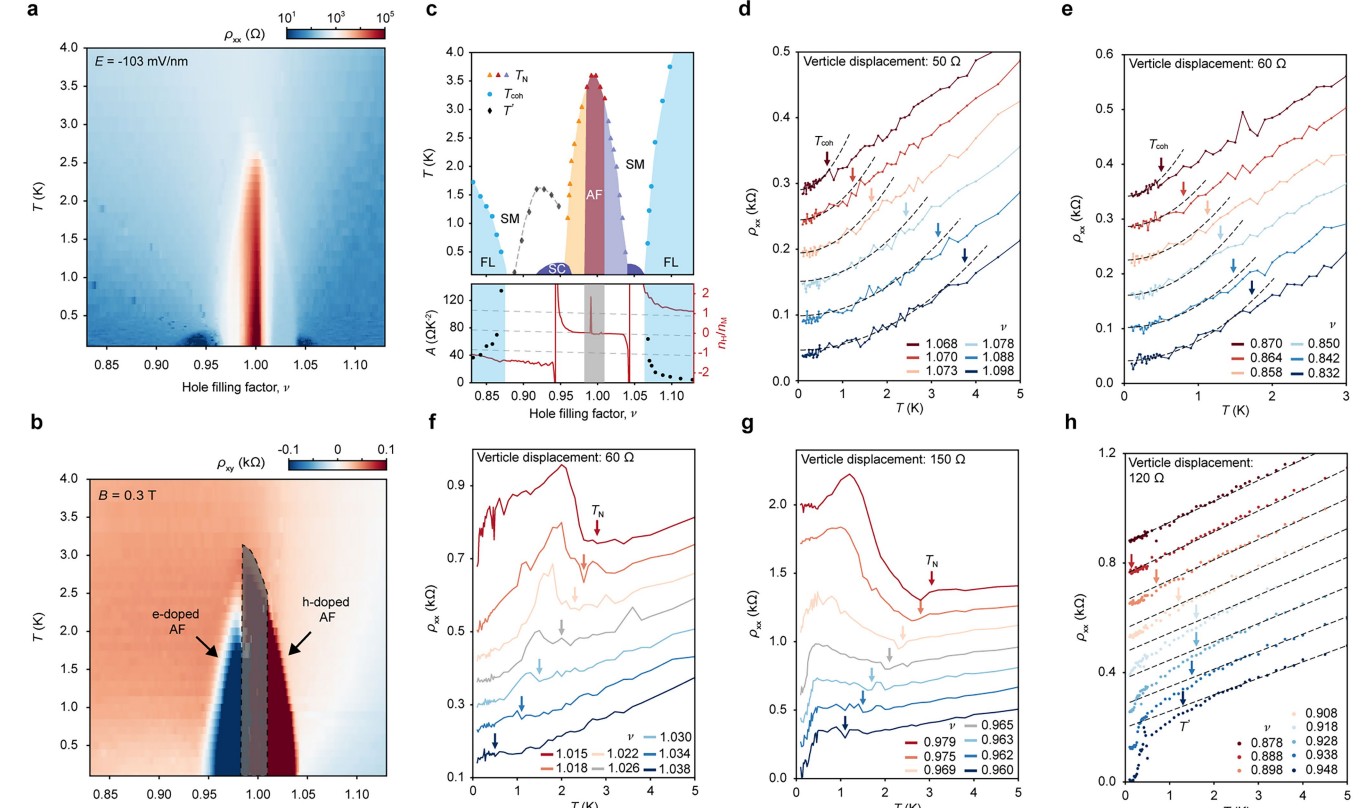

**Extended Data Fig. 6 | Analysis of DC transport at $E = -103$ mV/nm.**
**a,b**, Resistivity at $B = 0$ T (**a**) and Hall resistivity probed at $B = 0.3$ T (**b**) versus $\nu$
and $T$. The grey-shaded region in **b** is the AF insulator, where $\rho_{xy}$ cannot be
reliably measured. The electron-doped and hole-doped AF insulators have low
Hall densities; they develop only when the AF insulator is stabilized below $T_N \approx$
3.5 K. **c**, Upper panel: $\nu$ - $T$ phase diagram extracted from **a,b** (copy of Fig. 3e).
Lower panel: filling factor dependence of coefficient $A$ (left axis) and Hall
density (right axis) at $T = 50$ mK (copy of Fig. 3i). **d-h**, $T$-dependence of $\rho_{xx}$ at

representative fillings corresponding to the data points in **c**. The curves are
vertically displaced for clarity. The black dashed lines in **d,e** are fits to the
low-temperature part of the data using $\rho_{xx} = \rho_0 + AT^2$. The black dashed lines in **h**
are linear fits to the high-temperature part of the data. We estimate $T_{coh}$ and $T'$
using 10% deviation of the data from the fit; we estimate $T_N$ from the local
resistivity minimum. The estimated temperature scales are marked by arrows.
Data collected from device 1.

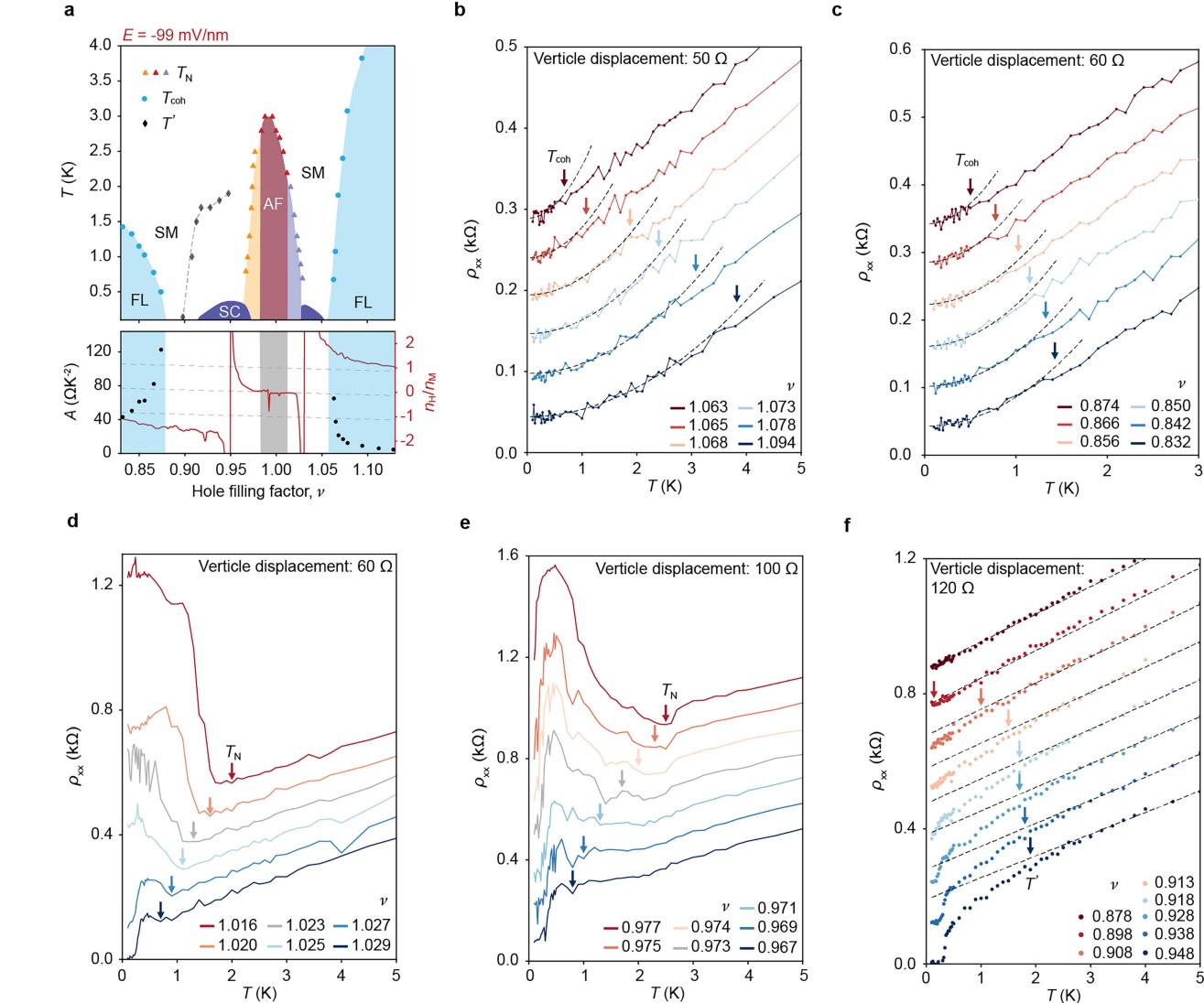

**Extended Data Fig. 7 | Analysis of DC transport at *E* = −99 mV/nm. a**, Copies of Fig. 3f,j of the main text. **b-f**, *T*-dependence of $\rho_{xx}$ at representative fillings corresponding to the data points in **a**. Same notation and convention as in Extended Data Fig. 6.

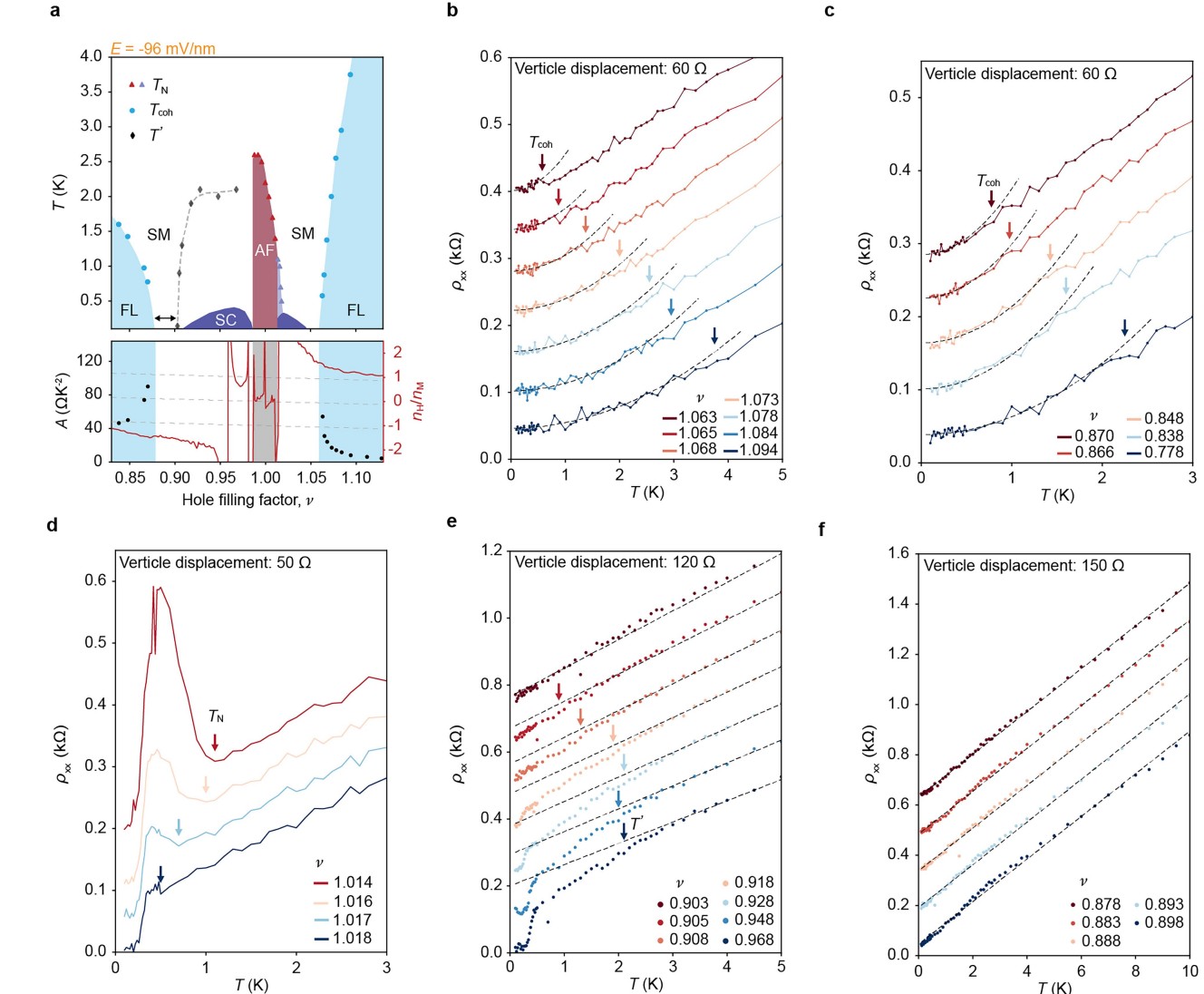

**Extended Data Fig. 8 | Analysis of DC transport at $E = -96$ mV/nm. a**, Copies of Fig. 3g,k of the main text. **b-e**, $T$-dependence of $\rho_{xx}$ at representative fillings corresponding to the data points in **a**. **f**, Linecuts in the strange metal region. The black dashed lines are linear fits to the data over the entire temperature range. Same notation and convention as in Extended Data Fig. 6.

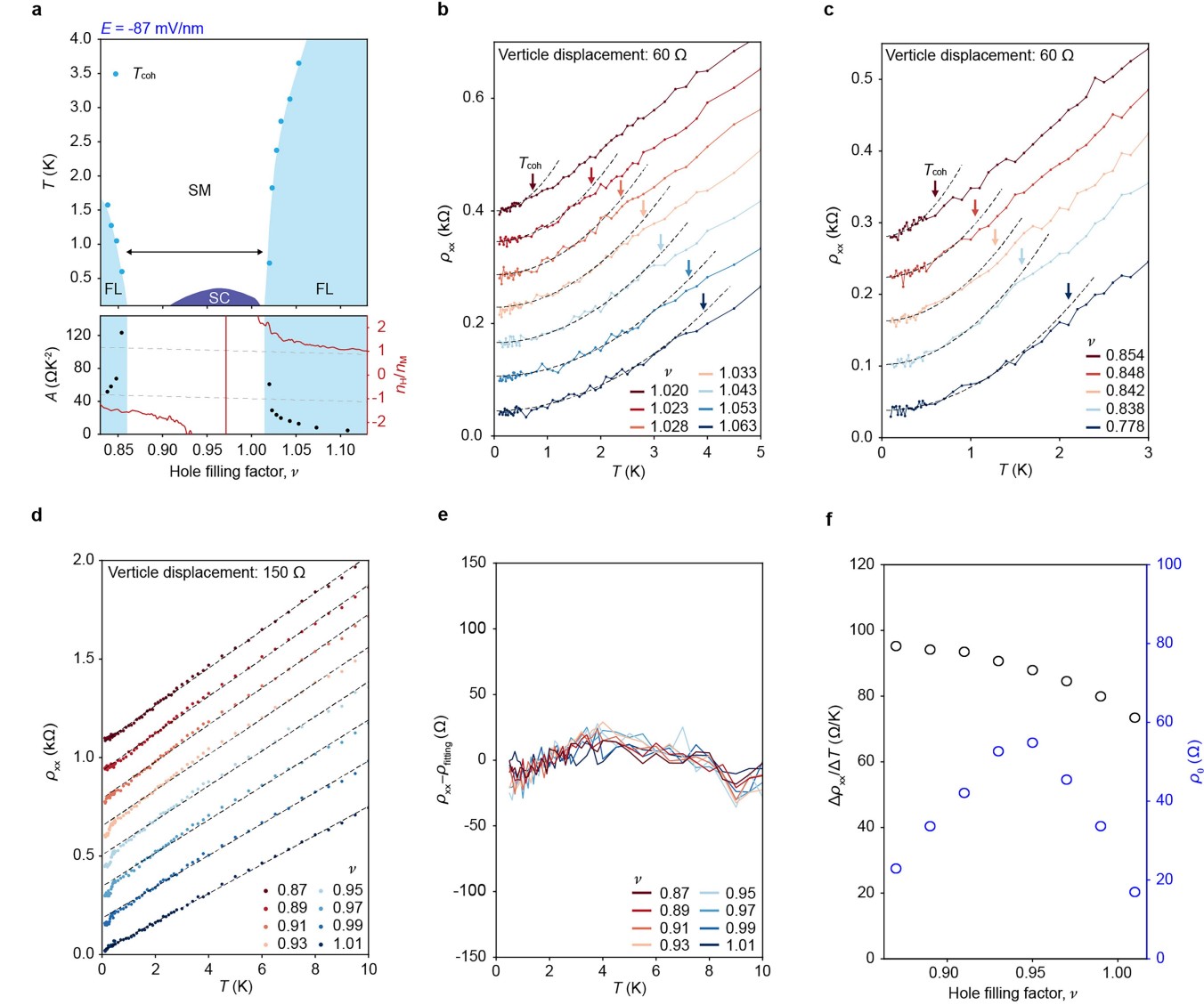

**Extended Data Fig. 9 | Analysis of DC transport at $E = -87\,mV/nm$. a**, Copies of Fig. 3h,l of the main text. **b,c**, $T$-dependence of $\rho_{xx}$ at representative fillings corresponding to the blue data points in **a**. **d**, Linecuts in the SM (strange metal) region. **e**, Residual of the linear fits in **d**. **f**, Filling factor dependence of the slope (left axis, black symbols) and the residual resistivity (right axis, blue symbols) for data in **d**. Same notation and convention as in Extended Data Fig. 6.

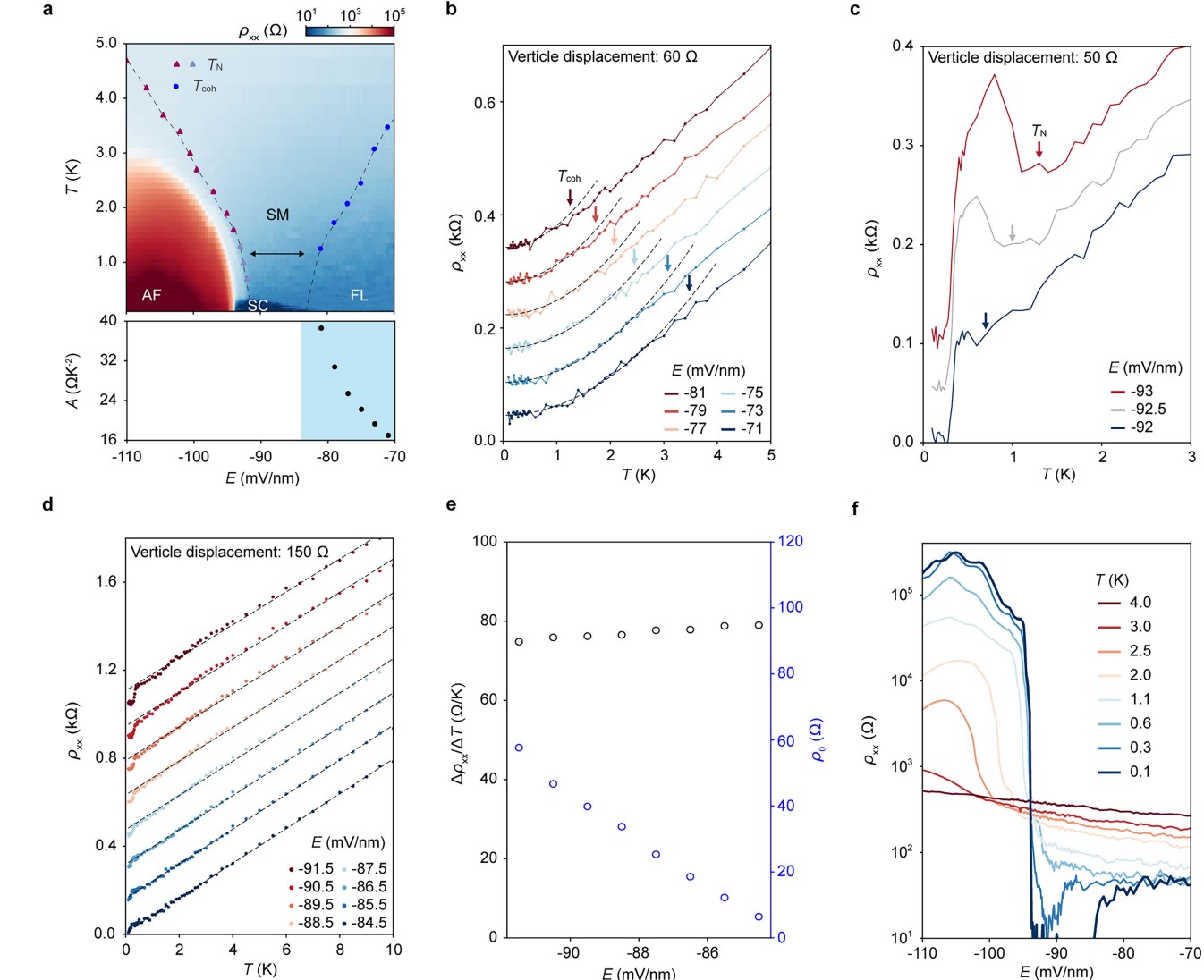

**Extended Data Fig. 10 | Analysis of DC transport at ν = 1. a**, Upper panel: $\rho_{xx}$ versus $E$ and $T$ at $B = 0$ T. Lower panel: coefficient $A$ versus $E$ for the Fermi liquid phase. As Mott transition is approached from the insulator side, $T_N$ decreases continuously and vanishes, a superconducting dome emerges immediately. AF metal is present next to the AF insulator near the critical $E$-field for Mott transition for $T_c < T < T_N$. **b,c**, $T$-dependence of $\rho_{xx}$ at representative $E$-fields. **d**, Linecuts of **a** at selected $E$-fields in the strange metal region. Dashed lines are linear fits to data (symbols) in the entire temperature range. **e**, $E$-field dependence of the slope (left axis, black symbols) and the residual resistivity (right axis, blue symbols) for data in **d. f**, Linecuts of **a** at selected temperatures. At 50 mK, $\rho_{xx}$ changes by more than four orders of magnitude in a narrow window of $E$-field (≈1 mV/nm) centered around $E = -94$ mV/nm, suggesting a direct transition from an AF insulator to a superconductor. Same notation and convention as in Extended Data Fig. 6.

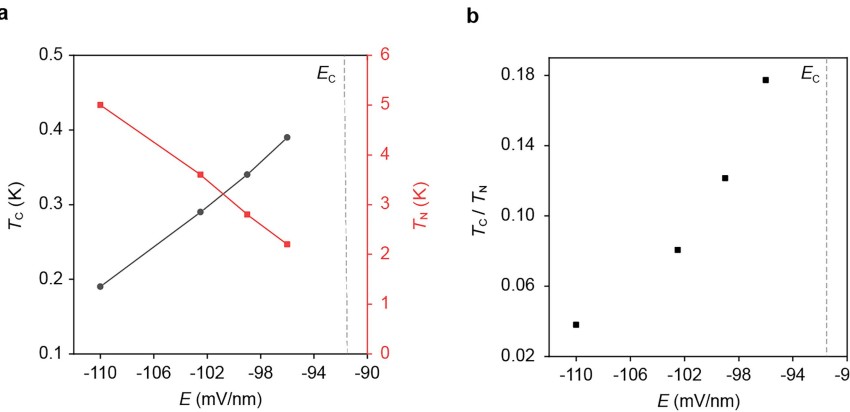

**a**

**b**

**Extended Data Fig. 11 | E-field dependence of $T_c/T_N$. a**, Optimal $T_c$ (left axis, black) and $T_N$ (right axis, red) for the AF insulator as a function of E-field at B = 0 T. The optimal $T_c$ at a given E-field is the highest $T_c$ from the $v$ - $T$ phase diagram in Fig. 3. Grey dashed line: critical E-field for the Mott transition. **b**, $T_c/T_N$ as a function of E-field; the ratio increases substantially near $E_c$. Data collected from device 1.

**a**

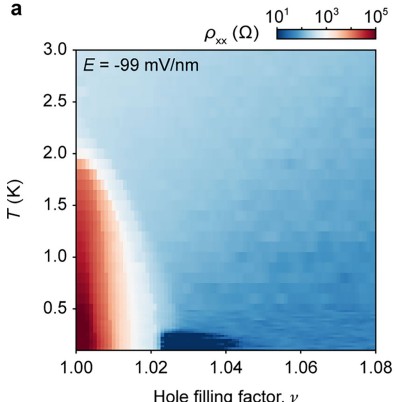

**b**

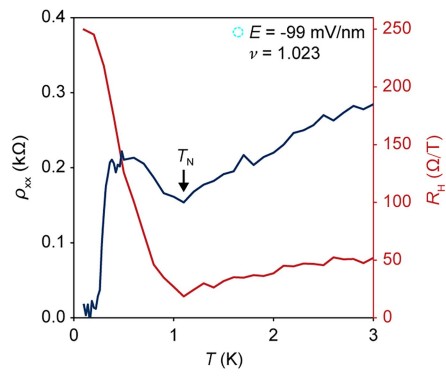

**Extended Data Fig. 12 | Coexistence of antiferromagnetism and superconductivity ($E = -99$ mV/nm). a**, $\rho_{xx}$ versus $\nu$ and $T$ at $B = 0$ T. **b**, $T$-dependence of $\rho_{xx}$ (left axis, blue) and Hall coefficient $R_H$ (right axis, red) at $\nu = 1.023$, where the superconducting dome in **a** overlaps with the AF metal.

AF metal with a small Fermi surface emerges below $T_N$, manifesting a bump in $\rho_{xx}$ and a jump in $R_H$. The AF metal turns into a superconductor at $T_c \approx 280$ mK. Data collected from device 1.