## [Peer Review file · Nature]

Bandwidth-tuned Mott transition and superconductivity in moiré WSe₂

Corresponding Author: Dr Kin Fai Mak

Version 0:

Reviewer comments:

Referee #1

(Remarks to the Author)

Referee Report on "Simulating high-temperature superconductivity in moiré WSe₂" (Nature ms 2025-06-15935)

Summary

This manuscript presents an experimental investigation of superconductivity in 4.6° twisted bilayer WSe₂ (tWSe₂), focusing on its moderate interaction regime, where $U/t \sim 1$. The authors explore the electronic phase diagram using a dual-gated device architecture, varying temperature, filling, and perpendicular electric field. They observe a superconducting dome adjacent to correlated insulating states and report signatures of antiferromagnetic (AF) order via magnetic circular dichroism (MCD). A central claim is that the observed phase diagram recapitulates key features of high-T_c cuprates, positioning tWSe₂ as a tunable platform to simulate high-T_c superconductivity.

While the study adds data from a previously unexplored twist angle, I do not find the results to rise to the level of novelty and conceptual advance required for Nature. The key conclusions largely reiterate findings from earlier works, particularly Ref. [11], with only incremental extensions. Below, I provide detailed comments.

Comments

1. Incremental advance over previous studies

The reported phase diagram and physical properties (e.g., T_c, coherence length, critical fields) closely resemble those in Ref. [11] (5° tWSe₂). The main distinction is the emergence of an AF insulating state at $\nu = 1$, embedded within the AF metallic phase. However, this variation likely stems from modest differences in twist angle or screening environment, not from a fundamentally new mechanism. The superconductivity itself displays no qualitatively new features relative to prior work.

2. Insufficient evidence for a direct SC–AF insulator transition

The authors suggest the possibility of a direct superconducting to antiferromagnetic insulator transition. From Fig. 2b, this claim is ambiguous. The data do not clearly rule out the presence of an intermediate metallic phase. The transition appears continuous and lacks evidence of a phase boundary directly connecting SC and AF insulator without an intervening AF metal. Additional data (e.g., resistivity or susceptibility cuts at fixed filling and varying electric field) are needed to support or falsify this scenario.

3. Lack of systematic angle-dependent study

A key shortcoming is the absence of a systematic study across different twist angles under comparable screening

conditions, given the known sensitivity of the correlated phases in moiré systems to twist angle and screening environment. The authors include low-temperature resistivity maps for other twist angles (Fig. 1d) but do not discuss superconducting behavior in these devices. A comparative study of SC and AF phase boundaries as a function of twist angle (under matched dielectric and gating conditions) would substantially strengthen the manuscript.

4. Questionable estimation of T_c / T_F

The claim of tightly bound Cooper pairs is central to the manuscript's analogy with high- T_c cuprates. However, the authors' estimation of T_F is based on a parabolic band approximation, which is clearly invalid near van Hove singularities and in the presence of strong correlation and Fermi surface reconstruction. This simplification leads to an overestimated $T_c / T_F \sim 0.06$. In contrast, Ref. [11] finds $T_c / T_F \sim 0.0008$ in a nearly identical system. Moreover, the reported coherence length $\xi / a_M \sim 10$ indicates weak coupling. The conclusion of tightly bound pairs is therefore not justified and should be re-evaluated with a more realistic band structure or experimental estimate of T_F .

5. MCD measurements confirm expected AF order but are not surprising

The MCD measurements demonstrate AF order at $\nu = 1$ and its persistence under light doping. While useful as a consistency check, this result is not unexpected in this system and does not, in my view, constitute a major conceptual advance.

6. Interpretation of Hall density jump

The Hall density jump across the correlated insulator is presented as support for similarities with cuprates. However, such features have been observed in various moiré systems and are not necessarily exclusive to high- T_c materials so far. Without additional evidence, the connection between the Hall anomaly and superconductivity remains speculative.

Conclusion

This work contributes additional data on superconductivity in $tWSe_2$, extending prior studies to a slightly different twist angle. However, its central findings are broadly consistent with existing literature, particularly Ref. [11], and do not, in my assessment, constitute a sufficiently novel or conceptual advance for publication in Nature. A more systematic exploration of twist-angle dependence and a more rigorous analysis of the superconducting state are needed to elevate the significance of the results. Therefore, I do not recommend publication in Nature in its current form.

Referee #2

(Remarks to the Author)

This manuscript presents a detailed exploration of superconductivity and correlated phases in twisted bilayer WSe_2 at twist angles between 3.6° and 5° , focusing especially on 4.6° . It maps out a series of phase diagrams reminiscent of high- T_c superconductors, including an antiferromagnetic insulating phase, superconducting domes, and unusual metallic behaviors such as "strange metal" states. Notable contributions include the direct observation of antiferromagnetic order via magnetic susceptibility measurements and a careful exploration of how these correlated states evolve with doping and electric field tuning. However, some interpretations and claims, such as the existence of a pseudogap and Planckian dissipation, require further clarification and support.

Comments and questions for the authors:

1. The title describes the system as "simulating high-temperature superconductivity." Could you clarify whether, or in which aspects, the system is genuinely a simulation rather than a distinct, correlated system with some phenomenological similarities and notable differences compared to cuprates? In what ways are the lattice structure and bands comparable to high- T_c superconductors?
2. Furthermore, twisted TMD structures are known to possess topological properties that distinguish them from many high- T_c materials. Can you elaborate on how the differences in topology, Berry curvature, etc affect comparisons with high- T_c superconductors?
3. Is there a clear reason why the 4.6° device exhibits clear cuprate-like phases whereas the 4.9° device does not, despite previous reports (e.g., Guo...Dean et al.) showing superconductivity at 5° ?
4. One of the major benefits of this system relative to high- T_c materials is its tunability. What novel insights do we gain from the tunability of the moiré phase diagram that are not accessible in conventional high- T_c materials?
5. Is there sufficient evidence to designate the observed unusual metallic state as a "strange metal" rather than merely linear- T resistivity? Could electron-phonon scattering be a contributing factor, especially considering that the Debye temperature might be suppressed in moiré systems with reduced carrier density (see e.g. Hwang PRL 99, 2019)?
6. In particular, you have not compared the observed T^* to the Debye temperature explicitly. Could you include this

comparison or justify its omission?

7. Given that direct spectroscopic evidence of a pseudogap is absent, might it be more appropriate to describe the observed behavior as a sub-linear resistivity region rather than a "possible pseudogap" phase?

8. In Line 221, you mention the system "possibly" exhibiting Planckian dissipation. Could you clarify whether your measured proportionality constant α (~ 1) is an experimentally extracted value or a theoretical expectation? Additionally, providing more precise numerical values and error bars for α would strengthen your claims. For that matter, when I plug in the moiré density, their effective mass, and T-linear slope I obtain a value for alpha between 7 to 9, not 1. Please explain the choices you made when evaluating this figure more clearly.

9. In discussing tightly bound Cooper pairs based on moiré period and coherence lengths, why not use the experimentally measured Hall density, especially since the presence of the AFM insulator significantly modifies the simple estimate?

10. Lines 140 and 170 report two different Néel temperatures. Could you clarify whether these figures correspond to different devices or measurement conditions?

Addressing these points would significantly enhance the clarity and impact of the manuscript. I believe that with these points addressed, it would be of interest to both the moiré materials and high-T_c communities, and to researchers interested in 2D materials and strongly correlated materials more generally.

Referee #3

(Remarks to the Author)

I co-reviewed this manuscript with one of the reviewers who provided the listed reports.

Referee #4

(Remarks to the Author)

See attached report.

Version 1:

Reviewer comments:

Referee #1

(Remarks to the Author)

The authors have satisfactorily addressed all of my questions and resolved the concerns I previously had regarding this work. After reviewing the revised manuscript and the authors' responses to the other two referees, I now support the publication of this manuscript in Nature.

Optional further question:

The data presented in Figure 1(e–g) of the revised manuscript are quite compelling. In Response 3, the authors state: "We expect T_c to drop quickly at large twist angles because the moiré effects are washed out and WSe₂ without moiré effects is not a superconductor under our experimental conditions." This raises a natural question: Have the authors performed measurements on samples with larger twist angles, specifically beyond 5 deg? If not, is there a particular reason—such as fabrication limitations or sample quality—that restricts the experimental exploration to twist angles below this threshold?

Referee #2

(Remarks to the Author)

The authors have largely addressed my concerns. In my opinion, the additions and modifications to the manuscript have significantly improved it. The twist-angle dependence of the low-temperature phase diagram in particular is a valuable addition and strengthens their argument that this system enables bandwidth tuning not available in high-T_c systems.

I have only one major concern remaining (more of a recommendation perhaps) and two minor ones. My major concern is the title. I still do not feel it is appropriate to describe this system as a "simulation" of high-temperature superconductors. As the authors themselves note in their rebuttal, if the work simulates anything, it is the triangular-lattice Hubbard model. They can tune the bandwidth and investigate how this modifies the phase diagram. I agree that this is a compelling contribution to the study of Hubbard lattices and that it may shed light on the behavior of more complicated systems such as high-temperature superconductors. Considering this, I think the title should reflect the true novelty of their work and avoid overstating (even unintentionally) the connection to high-temperature superconductors. For instance, titles like "Programmable triangular-lattice Hubbard physics in twisted WSe₂", or "Bandwidth-tuned Mott transition and superconductivity in twisted WSe₂" would keep the focus where it belongs, on the novel tunability afforded by twist angle and electric-field control of the bandwidth.

My two minor concerns are as follows:

1. The revised Fig. 1 is significantly improved over the original by the lower-temperature resistance maps and twist-angle

analysis, but the twist angles do not exactly match those in the original Fig. 1. Did the twist-angle estimates for each of these devices change, or are these measurements from new devices?

2. The "overall trend" of T_{BKT} versus twist suggested by the shaded region and the dashed line seems insufficiently justified. The data appear to show a cluster around 200 mK, two devices near 350 mK, and two that do not superconduct. That information is valuable on its own, but the trend is not clear enough to highlight based on these points alone. Otherwise, are the authors indicating that they expect T_{BKT} to continue increasing beyond 5 degrees, as the trend line implies?

Overall, I find the revisions thorough and the results compelling. I believe the results are high impact and will be of broad interest, and are thus suitable for publication in Nature.

Referee #3

(Remarks to the Author)

I co-reviewed this manuscript with one of the reviewers who provided the listed reports.

Referee #4

(Remarks to the Author)

See attached report.

Referee #1 (Remarks to the Author):

Referee Report on “Simulating high-temperature superconductivity in moiré WSe₂” (Nature ms 2025-06-15935)

Summary

This manuscript presents an experimental investigation of superconductivity in 4.6° twisted bilayer WSe₂ (tWSe₂), focusing on its moderate interaction regime, where $U/t \sim 1$. The authors explore the electronic phase diagram using a dual-gated device architecture, varying temperature, filling, and perpendicular electric field. They observe a superconducting dome adjacent to correlated insulating states and report signatures of antiferromagnetic (AF) order via magnetic circular dichroism (MCD). A central claim is that the observed phase diagram recapitulates key features of high- T_c cuprates, positioning tWSe₂ as a tunable platform to simulate high- T_c superconductivity.

While the study adds data from a previously unexplored twist angle, I do not find the results to rise to the level of novelty and conceptual advance required for Nature. The key conclusions largely reiterate findings from earlier works, particularly Ref. [11], with only incremental extensions. Below, I provide detailed comments.

Comments

1. Incremental advance over previous studies

The reported phase diagram and physical properties (e.g., T_c , coherence length, critical fields) closely resemble those in Ref. [11] (5° tWSe₂). The main distinction is the emergence of an AF insulating state at $\nu = 1$, embedded within the AF metallic phase. However, this variation likely stems from modest differences in twist angle or screening environment, not from a fundamentally new mechanism. The superconductivity itself displays no qualitatively new features relative to prior work.

Response 1:

We thank the reviewer for taking the time and effort necessary to review our manuscript. We appreciate all valuable comments and suggestions, which helped us to improve the manuscript. The focus of this study is not on the superconducting state, which, as the reviewer pointed out, may not differ fundamentally from the earlier reports. As we discuss in detail below, the study focuses on new aspects, especially those related to the normal states. Together with superconductivity, they demonstrate in tWSe₂ for the first time a phase diagram reminiscent of the high- T_c phase diagram. The study offers important new insights into the high- T_c problem. It

establishes the highly tunable tWSe2 system as a simulator of superconductivity in the triangular lattice Hubbard model. Because of these reasons, we disagree with the reviewer's assessment of the work as being incremental.

Below we first list the major new observations compared to Ref. 11:

1. An antiferromagnetic insulator at $\nu = 1$. This is supported by combined electrical transport and magneto-optical measurements.
2. Survival of the antiferromagnetic order upon doping the $\nu = 1$ insulator. This is revealed by the evolution of the susceptibility bump with doping (Fig. 2f and Extended Data Fig. 5c) and by the reconstructed Fermi surface (Fig. 4b,c and Extended Data Fig. 6).
3. Emergence of superconducting domes upon quenching antiferromagnetism by doping and electric field.
4. Strange metal behavior over two orders of magnitude in temperature (down to the lowest temperature) and over a range of doping density and perpendicular electric field. Note that the strange metal behavior is not only revealed by the T-linear ρ_{xx} , but also by a T²-dependence of the inverse Hall angle (Fig. 4). The latter is inconsistent with a quasiparticle description of electrical transport (see **Response 12**). The strange metal phase is also in the clean limit with residual resistivity down to $< 10 \Omega$. By contrast, twisted bilayer graphene and many cuprates show nearly two orders of magnitude larger residual resistivity.
5. Continuous evolution of the strange metal behavior to a conventional Fermi liquid behavior with doping density and electric field.
6. Hall density jumps at the peak of the superconducting domes (see **Response 6**).
7. Evolution of the ground state phase diagram with twist angle (Fig. R1). This is motivated by the reviewer's comment (see **Response 3**).

These new observations combined demonstrate for the first time in tWSe2 a phase diagram reminiscent of the high-Tc phase diagram. Next, we discuss the new insights that were brought by these results into the high-Tc problem. They were achieved thanks to the relatively simple electronic structure, highly tunable electronic properties, and relatively low energy scales of tWSe2.

Moiré WSe2 has a much simpler electronic structure near the Fermi level than cuprates and many other high-Tc materials. It has been shown that tWSe2 can be well described by the Hubbard model on a triangular lattice [e.g. Physical Review Research 2, 033087 (2020), Physical Review B 104, 075150 (2021), Nature Communications 12, 642 (2021) and Physical Review B 111, 014507 (2025)]. Our observation of high-Tc phenomenology in tWSe2 suggests that a simple Hubbard model could capture the essential physics of high-Tc superconductors. This is a significant point of debate in cuprates.

The ability to access nearly the entire superconducting phase diagram in a single tWSe2 sample by gating removes many complications or extrinsic effects (such as disorders) that cannot be

avoided in cuprates and other high-T_c materials which rely on synthesis of many samples to access the entire phase diagram. We also examined how the phase diagram evolves with perpendicular electric field and twist angle, which effectively tune the electronic bandwidth (see **Response 3**). Our results show that the highest T_c is achieved right next to the Mott transition, where the antiferromagnetic insulator is quenched by the electric field. The twist angle dependence of the ground state phase diagram (revised Fig. 1) further confirms that the superconducting state closely follows the Mott transition rather than the location of the van Hove singularity. These observations strongly suggest that the superconducting state has its root in electronic correlations rather than pure electron-phonon coupling (otherwise, the most robust superconducting state would appear at the van Hove singularity).

The relatively low energy scales of moiré systems allow us to access the physics of higher effective energy scales which is impossible or difficult in high-T_c materials. This helps to shed new light on the normal states of the superconductors. For instance, we were able to thermally quench the antiferromagnetic insulator because of the low Mott/antiferromagnetic scale in tWSe₂ (Fig. 3). We were also able to quench the superconducting state using a small magnetic field (< 0.2 T) because of the low absolute T_c (although the relative T_c is high) in tWSe₂ (Extended Data Fig. 4). By contrast, a very high magnetic field is required to quench superconductivity in high-T_c materials, which often induces other phase transitions or complications. The low upper critical field in tWSe₂ allowed us to access the strange metal phase down to the lowest temperature. We have verified that the strange metal phase exists over a range of doping densities rather than a single critical density or critical point (revised Fig. 4). This is another open question in high-T_c.

In summary, we have revised our manuscript by incorporating some of the above discussions to bring to the fore the important advances achieved in this work (changes highlighted). We have also revised Fig. 1 to include the twist angle dependent ground state phase diagram.

Figure R1. Twist angle effect. **a**, Longitudinal resistance R versus ν and E at $T = 50$ mK and $B = 0$ T for twist angles ranging from 2.1° to 4.7° . The correlation effect weakens with increasing twist angle. Superconductivity is observed for twist angle larger than about 3.5° . The dashed line marks the location of the ν Hs in the 4.0° sample. **b**, Extracted $\theta - E$ phase diagram at $\nu = 1$ and $T = 50$ mK exhibiting a correlated insulator (red), superconductor (SC, dark blue) and an AF metal (yellow). The dashed line separates the layer-hybridized and layer-polarized regions. Data points are extracted from devices with different θ 's; they guide the construction of the phase boundaries. **c**, Superconducting transitions (at optimal T_c) for devices with different θ 's. **d**, The extracted optimal T_{BKT} (the Berezinskii-Kosterlitz-Thouless transition temperature) as a function of θ . The shaded area represents an overall trend for T_{BKT} versus θ .

2. Insufficient evidence for a direct SC–AF insulator transition

The authors suggest the possibility of a direct superconducting to antiferromagnetic insulator transition. From Fig. 2b, this claim is ambiguous. The data do not clearly rule out the presence of an intermediate metallic phase. The transition appears continuous and lacks evidence of a phase boundary directly connecting SC and AF insulator without an intervening AF metal. Additional data (e.g., resistivity or susceptibility cuts at fixed filling and varying electric field) are needed to support or falsify this scenario.

Response 2:

Additional data are included in revised Extended Data Fig. 10 (also Fig. R2) to support an insulator-to-superconductor transition induced by a perpendicular electric field at fixed $\nu = 1$. Figure R2a shows ρ_{xx} versus E and T at $\nu = 1$. Fig. R2b and R2c show several temperature and electric-field linecuts, respectively. At low temperatures (e.g. 0.1K), ρ_{xx} shows a more than 4 orders of magnitude jump for a very small change in the electric field (by about 1 mV/nm). On the left, the system is an insulator, and on the right, a superconductor. To the best of our knowledge, this is the sharpest non-thermal transition from an insulator to a superconductor among all the reported ones. The small but finite electric-field width of the transition is presumably caused by disorders in the sample.

Regarding a possible intervening antiferromagnetic metal, we note that the electric-field linecuts in Fig. R2c indeed show the presence of such a metallic state. This is revealed by a ρ_{xx} bump below T_N . However, the state is unstable towards a superconducting transition at $T_c < T_N$, giving rise to a sharp insulator-to-superconductor transition at low temperatures.

To summarize, the sharp electric-field-induced transition at low temperatures shows a clear phase boundary between an insulator and a superconductor. There is an antiferromagnetic metal immediately adjacent to the insulator for $T > T_c$ but this metallic state becomes a superconductor at low temperatures. We have provided additional linecuts in the revised Extended Data Fig. 10.

Figure R2. Electric-field-induced superconductor-to-insulator transition. **a**, Longitudinal resistivity ρ_{xx} as a function of electric field E and T at $\nu = 1$. The black arrow marks the antiferromagnetic metal phase (with higher ρ_{xx}) right next to the insulator. **b**, Linecuts of **a** at $T = 0.1$ K – 4.0 K. At $T = 0.1$ K, ρ_{xx} changes by more than 4 orders of magnitude in a narrow window (≈ 1 mV/nm) centered around $E = -94$ mV/nm. **c**, Linecuts of **a** at $E = -93$ mV/nm, -92.5 mV/nm and -92 mV/nm. An AF metal develops below T_N (denoted by an arrow for $E = -93$ mV/nm) and turns into a superconductor below $T_c < T_N$.

3. Lack of systematic angle-dependent study

A key shortcoming is the absence of a systematic study across different twist angles under comparable screening conditions, given the known sensitivity of the correlated phases in moiré systems to twist angle and screening environment. The authors include low-temperature resistivity maps for other twist angles (Fig. 1d) but do not discuss superconducting behavior in these devices. A comparative study of SC and AF phase boundaries as a function of twist angle (under matched dielectric and gating conditions) would substantially strengthen the manuscript.

Response 3:

We thank the reviewer for the suggestion. We took this suggestion seriously and studied the twist angle dependence of the ground state phase diagram under similar screening conditions (i.e. similar hBN dielectric thicknesses). Figure R1 shows ρ_{xx} versus ν and E (at $T = 50$ mK) for samples with different twist angles. The superconducting state follows the Mott transition (induced by E-field) rather than the van Hove singularity. The observation strongly suggests that the superconducting state has its root in Mott physics and antiferromagnetism rather than pure electron-phonon coupling (otherwise, the most robust superconducting state would appear at the van Hove singularity).

The shift of the $\nu = 1$ insulator to higher electric fields with increasing twist angle can be understood in the following way. The electric field tunes the band structure including the location of the van Hove singularity (vHS), and the doping density tunes the Fermi level (Fig. 1b,c). The closer the singularity is to the Fermi level, the lower is the Fermi velocity and the more stable is the correlated insulator at $\nu = 1$. This picture largely explains the absence of the $\nu = 1$ insulator at both small and large electric fields in samples with intermediate twist angles ($\theta \approx 3.6^\circ - 4.6^\circ$) because the vHS is far away from the Fermi level in both limits; only at intermediate electric fields the vHS is close enough to the Fermi level to stabilize the $\nu = 1$ insulator. For $\theta \gtrsim 4.7^\circ$, the correlations are too weak to stabilize the insulator even with the help of the vHS; and for $\theta \lesssim 3.5^\circ$, the insulator is always stabilized in the layer-hybridized region due to the narrow moiré bandwidth regardless of the location of the singularity. Thus, both θ and E can effectively tune the band flatness at the Fermi level and realize a bandwidth-tuned Mott transition at $\nu = 1$. Our result in Fig. R1a shows that the most robust superconducting state always appears right next to the Mott transition at lower electric fields.

To further illustrate the relationship between the antiferromagnetic insulator at $\nu = 1$ and the superconducting state, we show a twist angle-electric field phase diagram at $\nu = 1$ in Fig. R1b. It was constructed using a total of 8 samples (data points). The phase diagram further demonstrates that the superconducting state always appears right adjacent to the bandwidth-tuned Mott transition.

Finally, we also show the twist angle dependence of the optimal T_c (for the superconducting state right next to the $\nu = 1$ insulator) in Fig. R1c,d. The optimal T_{BKT} increases with twist angle. We expect T_c to drop quickly at large twist angles because the moiré effects are washed out and WSe2 without the moiré effects is not a superconductor under our experimental conditions.

In the revised manuscript, we have modified Fig. 1 to include the twist angle dependence. We have also included additional discussions on the twist angle dependence (page 3).

4. Questionable estimation of T_c / T_F

The claim of tightly bound Cooper pairs is central to the manuscript's analogy with high- T_c cuprates. However, the authors' estimation of T_F is based on a parabolic band approximation, which is clearly invalid near van Hove singularities and in the presence of strong correlation and Fermi surface reconstruction. This simplification leads to an overestimated $T_c / T_F \sim 0.06$. In contrast, Ref. [11] finds $T_c / T_F \sim 0.0008$ in a nearly identical system. Moreover, the reported coherence length $\xi / a_M \sim 10$ indicates weak coupling. The conclusion of tightly bound pairs is therefore not justified and should be re-evaluated with a more realistic band structure or experimental estimate of T_F .

Response 4:

We thank the reviewer for the comment. Before we elaborate on measurements/estimates, which support the scenario of tightly bound Cooper pairs in the under-doped region of our sample, we first comment on the seemingly very different $\frac{T_c}{T_F}$ in our work and in Ref. 11. For $\nu \approx 1.023$ in the under-doped region, we used the Hall density $0.023 n_M$ (which is about 40 times smaller than the moiré density n_M) to estimate T_F . We would have obtained a comparable $\frac{T_c}{T_F}$ (≈ 0.001) as in Ref. 11 if we had used the moiré density ($1.023 n_M$) to estimate T_F .

The typical ratio of the coherence length to the lattice period is $\frac{\xi}{a} \approx 5$ in cuprates [see Mourachkine, A. High-Temperature Superconductivity in Cuprates: The Nonlinear Mechanism and Tunneling Measurements (Kluwer Academic Publishers, 2002)]. By contrast, the typical ratio is $\frac{\xi}{a} \sim 100 - 10,000$ in weakly bound BCS superconductors. In general, Cooper pairs in superconductors with a coherence length a few times of the lattice scale are tightly bound. Therefore, our sample with $\frac{\xi}{a_M} \approx \frac{34}{4.1} \approx 8.3$ belongs to superconductors with tightly bound Cooper pairs. Here the moiré period plays the role of the lattice constant. We could also use the average inter-particle distance, which is $\sim 1/\sqrt{n_H}$, as a measure of length scale. For $\nu \approx 1.023$ in the under-doped superconducting region, superconductivity emerges from a small Fermi surface antiferromagnetic metal with density $n_H \approx 0.023 n_M$. This leads to a ratio of $\sqrt{n_H} \xi \sim 1.4$ (that is,

the size of the Cooper pairs is comparable to the average inter-particle distance), which again supports tightly bound Cooper pairs.

We also improved our estimate of $\frac{T_c}{T_F}$ in the under-doped region. We estimate $\frac{T_c}{T_F}$ for the superconductor denoted by a blue circle in the phase diagram (Fig. R3c, $\nu \approx 1.023$, $E = -99$ mV/nm), at which the superconducting dome intersects the antiferromagnetic metal (Fig. R3a). Figure R3b shows the T -dependence of ρ_{xx} and the Hall coefficient R_H . We observe an antiferromagnetic phase transition at $T_N \approx 1$ K, below which both ρ_{xx} and R_H increase due to Fermi surface reconstruction, followed by a superconducting phase transition at $T_c \approx 0.3$ K. The result confirms that the superconducting state develops from a small Fermi surface with density $(\nu - 1)n_M \approx 0.023 n_M$ ($n_M \approx 6.7 \times 10^{12} \text{cm}^{-2}$). Next, we extract the effective mass from the T -dependent Shubnikov-de Haas oscillations (Fig. R3e). However, clear quantum oscillations cannot be observed at a relatively low magnetic field at $E = -99$ mV/nm. At the same filling, quantum oscillations emerge when the electric field is tuned slightly away from $E = -99$ mV/nm. This indicates that the effective mass decreases. Therefore, we used the measured effective mass ($m^* \approx 0.5 m_0$) at $E = -105$ mV/nm (denoted by a pink circle in Fig. R3c) as an estimate of the lower bound of the effective mass. This allows us to estimate $T_F \approx \frac{\hbar^2 \pi * 0.023 n_M}{m^* k_B} \approx 8.5$ K and obtain $\frac{T_c}{T_F} \approx 3.5\%$ as a lower bound [the value is on par with our earlier work, Nature 637, 833–838 (2025), in which T_F can be directly obtained as the coherence temperature for electrical transport].

We agree with the reviewer that this estimate made use of the parabolic band approximation which could be subject to error for a reconstructed Fermi surface. However, given its consistency with earlier studies, we believe that our estimate $\frac{T_c}{T_F} \approx 3.5\%$ is reliable in the under-doped regime, where superconductivity develops from a small Fermi surface. On the other hand, the ratio $\frac{T_c}{T_F} \approx 0.001$ (using the full moiré density like in Ref. 11) is more accurate in the over-doped regime, where the Mott gap vanishes and the system has a large Fermi surface. In general, $\frac{T_c}{T_F}$ is doping dependent. The high $\frac{T_c}{T_F}$ ratio applies to the under-doped strongly correlated regime of the phase diagram.

Last, we note that we were able to determine the optimal $\frac{T_c}{T_N}$ from experiment. It varies from 4% to 18% for different electric fields (Extended Data Fig. 11). The values are comparable to that in the cuprates. This again shows that although the absolute T_c is low in moiré WSe₂, the material is a ‘high- T_c ’ superconductor.

In summary, all comparisons ($\frac{\xi}{a_M} \approx 8.3$, $\sqrt{n_H} \xi \approx 1.4$, $\frac{T_c}{T_F} \approx 3.5\%$ and $\frac{T_c}{T_N} \approx 4 - 18\%$) point to the emergence of high- T_c superconductivity and tightly bound Cooper pairs in the under-doped regime of the phase diagram. We have incorporated the above comparisons including the improved

estimate of $\frac{T_c}{T_F}$ in the revised Methods (page 9). We have also included Fig. R3 in Extended Data Fig. 2 and 13.

Figure R3. Estimation of T_c/T_F . **a**, ρ_{xx} versus ν and T at $E = -99\text{mV/nm}$ and $B = 0\text{T}$. **b**, T -dependence of ρ_{xx} (blue) and Hall coefficient R_H (red) at $\nu = 1.023$, where the superconducting dome in **a** overlaps with the AF metal. An AF metal with a small Fermi surface is observed below T_N , manifested by a bump in ρ_{xx} and a jump in R_H . The AF metal turns into a superconductor at $T_c \approx 280\text{mK}$. The result demonstrates that superconductivity in the under-doped region is developed from an AF metal with a small Fermi surface. **c**, ρ_{xx} versus ν and E at $T = 50\text{mK}$ and $B = 0\text{T}$. The blue (pink) circle marks where panel **b** (the effective mass m^*) was measured. **d**, ρ_{xx} versus ν and B measured along the black dashed line in **c**. A Landau fan emerges from $\nu = 1$, showing a small Fermi surface with carrier density $(\nu - 1)n_M$. **e**, Resistivity difference $\Delta\rho_{xx} = \rho_{xx}(T) - \rho_{xx}(1.4\text{K})$ as a function of ν at varying T and at $B = 2\text{T}$ (dashed line in **d**). **f**, T -dependence of the amplitude $|\Delta\rho_{xx}|$ at $\nu \approx 1.023$ as marked by the arrow in **e**. $|\Delta\rho_{xx}|$ was obtained as the difference between the nearest $\Delta\rho_{xx}$ peak and dip centered around ν . The quasiparticle effective mass m^* was extracted by fitting the data to $|\Delta\rho_{xx}| = \frac{R_a\lambda(T)}{\sinh\lambda(T)}$ (red lines).

Here R_a is the amplitude and $\lambda(T) = \frac{2\pi^2 k_B T m^*}{\hbar e B}$.

5. MCD measurements confirm expected AF order but are not surprising

The MCD measurements demonstrate AF order at $\nu = 1$ and its persistence under light doping. While useful as a consistency check, this result is not unexpected in this system and does not, in my view, constitute a major conceptual advance.

Response 5:

We agree with the reviewer that the MCD result does not constitute a major conceptual advance. It provides the first direct experimental demonstration of the AF order and its persistence in the under-doped region in the system. Combined with the transport results, it demonstrates the emergence of a high-T_c-like phase diagram in a tunable Hubbard model system. For these reasons, the MCD measurements present an important advance over previous studies.

6. Interpretation of Hall density jump

The Hall density jump across the correlated insulator is presented as support for similarities with cuprates. However, such features have been observed in various moiré systems and are not necessarily exclusive to high-T_c materials so far. Without additional evidence, the connection between the Hall anomaly and superconductivity remains speculative.

Response 6:

We thank the reviewer for the comment. Hall density jumps have been observed in other moiré materials (most notably in twisted graphene systems); they occur at Dirac revivals (near integer fillings) and/or the van Hove singularities (vHS) of the band structure. However, the observed Hall density jumps in this work are tied to the peaks of the superconducting domes (Fig. 3i-k). They are not related to the vHS in the band structure because 1) there are two jumps, one for each superconducting dome, with increasing doping density and 2) the jump size is one moiré density n_M (while a jump due to vHS is $2n_M$, see Fig. 3l). The Hall density jumps, which are observed only in the presence of an antiferromagnetic insulator in the phase diagram in this work, are thus distinct from what has been reported in other moiré materials. Combining with other observations, such as the strange metal phase, the crossover to the Fermi liquid in the over-doped regime and others, they demonstrate a global phase diagram like that in high-T_c superconductors. In the revised manuscript (page 9), we have included additional discussions on the observed Hall density jumps.

Conclusion

This work contributes additional data on superconductivity in tWSe₂, extending prior studies to a slightly different twist angle. However, its central findings are broadly consistent with existing

literature, particularly Ref. [11], and do not, in my assessment, constitute a sufficiently novel or conceptual advance for publication in Nature. A more systematic exploration of twist-angle dependence and a more rigorous analysis of the superconducting state are needed to elevate the significance of the results. Therefore, I do not recommend publication in Nature in its current form.

Response 7:

We hope that our responses above have sufficiently addressed the main criticism regarding the novelty of our work. Again, we thank the reviewer for the comments and suggestions which have helped us to improve our manuscript.

Referee #2 (Remarks to the Author):

This manuscript presents a detailed exploration of superconductivity and correlated phases in twisted bilayer WSe_2 at twist angles between 3.6° and 5° , focusing especially on 4.6° . It maps out a series of phase diagrams reminiscent of high- T_c superconductors, including an antiferromagnetic insulating phase, superconducting domes, and unusual metallic behaviors such as "strange metal" states. Notable contributions include the direct observation of antiferromagnetic order via magnetic susceptibility measurements and a careful exploration of how these correlated states evolve with doping and electric field tuning. However, some interpretations and claims, such as the existence of a pseudogap and Planckian dissipation, require further clarification and support.

Comments and questions for the authors:

1. The title describes the system as "simulating high-temperature superconductivity." Could you clarify whether, or in which aspects, the system is genuinely a simulation rather than a distinct, correlated system with some phenomenological similarities and notable differences compared to cuprates? In what ways are the lattice structure and bands comparable to high- T_c superconductors?

Response 8:

We thank the reviewer for the positive comments. We are also grateful for the suggestions which helped us to improve our manuscript. Our work attempts to simulate the complex phase diagram of 'high-temperature' superconductivity (high relative temperature) realized in the triangular lattice Hubbard model. We do not attempt to simulate the phase diagram of cuprates as both the lattice symmetry and the band structure are very different. The cuprates have a bipartite square lattice whereas our system has a non-bipartite triangular lattice, in which stronger kinetic frustration effects are expected. However, many of the correlated phases observed in the cuprates are observed in our system. These include an antiferromagnetic insulator at half-band-filling, antiferromagnetic metals (or spin density waves) in the under-doped regime, superconducting domes upon suppression of antiferromagnetism, a possible pseudogap (T -sublinear resistivity) phase, strange metals and Fermi liquids in the over-doped regime. The results suggest that much of the high- T_c phenomenology could be captured by the Hubbard model, which remains an important topic of debate in high- T_c .

In addition to simulating the high- T_c phase diagram induced by doping, we also examined how the phase diagram evolves with tuning the Fermi velocity near half-band-filling. The latter is tuned by a perpendicular electric field, which controls the energy of a van Hove singularity (see Fig. 1c). The closer the van Hove singularity is to the Fermi level, the lower is the Fermi velocity. The results show that the highest T_c is achieved right next to the Mott transition, at which the antiferromagnetic insulator is quenched by the electric field. Such electric-field or bandwidth tuning is not possible in the cuprates.

In the revised Fig. 1, we further demonstrated tuning of the bandwidth by twist angle. A sufficiently large twist angle (or equivalently, bandwidth) destroys the antiferromagnetic insulator at half-band-filling. Moreover, rather than following the location of the van Hove singularity, the superconducting state closely follows the Mott transition. The results strongly suggest that the observed superconductivity has its root in Mott physics and antiferromagnetism rather than pure electron-phonon coupling (otherwise, the most robust superconducting state would appear at the van Hove singularity). Again, such bandwidth tuning is not possible in the cuprates.

In summary, understanding cuprates and high-Tc physics is a motivation of our work, but we do not intend to simulate the cuprate phase diagram. Rather, we simulate the complex phase diagram of the triangular lattice Hubbard model and address the question whether the complex high-Tc phenomenology can emerge in a Hubbard model system. We have revised the manuscript accordingly to make this point clearer (changes are highlighted).

2. Furthermore, twisted TMD structures are known to possess topological properties that distinguish them from many high-Tc materials. Can you elaborate on how the differences in topology, Berry curvature, etc affect comparisons with high-Tc superconductors?

Response 9:

We thank the reviewer for the question. Unlike twisted MoTe₂, whose bands are topological over a wide range of twist angle, the topmost moiré valence band of twisted WSe₂ is not topological for twist angle above about 4 degrees [see Nature Physics 21, 1217–1223 (2025)]. This is supported by the absence of Chern insulators near $\nu = 1$ in our sample (4.6 degree) even under high magnetic fields [see Fig. R4 and arXiv:2509.19287]. By contrast, in tWSe₂ with twist angle less than about 3 degrees, the topmost moiré valence band is topological and a Chern insulating state at $\nu = 1$ has been reported [see Science 384, 343-347 (2024) and Nature Communications 16, 1959 (2025)]. Therefore, the band topology of twisted WSe₂ is twist angle dependent. The twist angle we have focused on in this study (4.6 degrees) does not support a topological band; the low-energy physics of the topmost moiré valence band in this limit can be mapped to the single-band Hubbard model on a triangular lattice [Physical Review Research 2, 033087 (2020), Physical Review B 104, 075150 (2021), Nature Communications 12, 642 (2021) and Physical Review B 111, 014507 (2025)].

Figure R4. Absence of a Chern insulating state at $\nu = 1$. Longitudinal resistance R as a function of ν and B at $E = -103$ mV/nm and $T = 1.5$ K. Chern insulator is not observed up to $B = 12$ T, which is consistent with the picture of a non-topological moiré band at $\theta = 4.6^\circ$.

On the other hand, the reviewer is correct that even without topological bands, strong Berry curvature effects could still be present in the topmost moiré valence band. This is especially the case under a finite perpendicular electric field, which is known to induce a spin splitting and Berry curvatures in the moiré band structure [Physical Review B 104, 075150 (2021)]. As shown in this theory paper, the effective Hamiltonian is a triangular lattice Hubbard Hamiltonian with an additional spin-orbit coupling term. The effect of this new term is to suppress the out-of-plane spin dynamics and turn the system into an effective XY model at half-band-filling in the strong coupling limit [e.g. Physical Review B 104, 075150 (2021) and Physical Review B 111, 014507 (2025)]. The new term therefore tends to stabilize a 120-degree Néel order at half-band-filling under finite perpendicular electric fields, which appears to be consistent with our experimental observation. How this new term affects the phase diagram away from half-band-filling is unclear and deserves further investigation.

In summary, we have focused on large twist angle samples in this study, for which the topmost moiré band is non-topological and can be mapped to a (modified) Hubbard Hamiltonian. Complications from non-trivial band topology are absent. In the revised manuscript (page 2), we have included a brief comment on the twist angle-dependent band topology in twisted WSe₂. We have also included Fig. R4 as Extended Data Fig. 12.

3. Is there a clear reason why the 4.6° device exhibits clear cuprate-like phases whereas the 4.9° device does not, despite previous reports (e.g., Guo...Dean et al.) showing superconductivity at 5°?

Response 10:

We thank the reviewer for the question. We believe that the reason a high- T_c like phase diagram is not observed in the study of Guo et al. is the large bandwidth or twist angle in their sample. We have performed additional measurements and the systematic angle dependence study in revised Fig. 1 clearly addresses this question. As twist angle increases, the bandwidth W increases, which quenches the correlated insulator at $\nu = 1$. Such bandwidth-tuned Mott transition is expected at $U/W \approx 1$ in a Hubbard model system, which corresponds to a critical twist angle about 4.7 degrees according to our results in Fig. 1. Without a Mott insulator, a high- T_c like phase diagram is not expected in the study of Guo et al.

The high- T_c phenomenology is believed to appear near a Mott transition in the Hubbard model and the complex phase diagram of high- T_c is expected to emerge upon doping a weak Mott insulator near the Mott transition [e.g. Reviews of Modern Physics 78, 17-85 (2006); Physical Review Letters 110, 216405 (2013)]. Our data (Fig. 1) seems to support this viewpoint: it shows that the superconducting state closely follows the destruction point of the Mott insulator. We have pointed out the twist angle-tuned Mott transition in the revised manuscript (page 3 and 8).

4. One of the major benefits of this system relative to high- T_c materials is its tunability. What novel insights do we gain from the tunability of the moiré phase diagram that are not accessible in conventional high- T_c materials?

Response 11:

We thank the reviewer for the question. The ability to tune the bandwidth (or the Fermi velocity) by the twist angle and the perpendicular electric field allows us to access the bandwidth-tuned Mott transition (continuously), which is not possible in bulk high- T_c materials. This enabled us to test whether high- T_c phenomenology is expected near a Mott transition, which remains a topic of debate in the community. Our observation of a superconducting state closely following the Mott transition (revised Fig. 1) supports this viewpoint. The observation also strongly suggests that the observed superconductivity has its root in Mott physics and antiferromagnetism rather than pure electron-phonon coupling. See **Response 8** and **10** for more details.

There are additional advantages brought by the tunability of the system. The ability to access nearly the entire phase diagram in a single sample by gating is superior to growing multiple samples to map out the phase diagram. It provides cleaner data with continuous control but also avoids complications from sample variations.

The low energy scales of moiré systems allow us to access the physics of higher effective energy scales which is impossible or difficult to access in high- T_c materials. This helps to shed new light on the parent states of the superconductors. For instance, we were able to thermally quench the antiferromagnetic insulator because of the low Mott/antiferromagnetic scale in tWSe₂ (Fig. 3). We

were also able to quench superconductivity using a small magnetic field (< 0.2 T) because of the low absolute T_c (although high relative T_c) in tWSe2 (Extended Data Fig. 4). By contrast, superconductivity can be quenched in high- T_c materials only by a very high magnetic field which often induces other phase transitions or complications. Particularly, the low upper critical field allowed us to access the strange metal phase down to the lowest possible temperature. As shown in our revised Fig. 4, we can now verify that the strange metal phase (after suppression of superconductivity) exists over a range of doping densities rather than a single critical density (or critical point). This is another open question in the high- T_c literature.

We have incorporated some of the above discussions in the revised manuscript.

5. Is there sufficient evidence to designate the observed unusual metallic state as a "strange metal" rather than merely linear-T resistivity? Could electron-phonon scattering be a contributing factor, especially considering that the Debye temperature might be suppressed in moiré systems with reduced carrier density (see e.g. Hwang PRL 99, 2019)?

Response 12:

We thank the reviewer for the question. Our observations cannot be explained by the electron-phonon coupling alone and support a ‘strange metal’ behavior as we discuss below.

First, the T-linear resistivity is observed down to about 100 mK and up to about 20 K in our experiment. The behavior spans over two orders of magnitude in temperature and, importantly, survives down to the lowest temperature in our measurement (revised Fig. 4 and Extended Data Fig. 6-10, superconductivity was quenched by an out-of-plane magnetic field). T-linear resistivity from electron-phonon coupling is expected for temperatures approximately above the Bloch-Gruneisen temperature $T_{BG} = \frac{2\hbar v_p k_F}{k_B}$. Following Phys. Rev. B 99, 165112 (2019), we estimated T_{BG} to be about 17.5 K for tWSe2 by using the phonon velocity $v_p \approx 2510$ m/s for WSe2 and the moiré density $n_M \approx 6.7 \times 10^{12}$ cm⁻² to estimate the Fermi wavevector $k_F \approx \sqrt{\pi n_M}$. Note that in our experiment, the strange metal behavior is observed near half-band-filling but outside the small Hall density regions of the phase diagram; the use of n_M to approximate k_F is therefore justified. The persistence of T-linear dependence down to temperatures about two orders of magnitude smaller than T_{BG} cannot be explained by the electron-phonon coupling.

Second, the T-linear behavior is accompanied by a T^2 -dependence of the inverse Hall angle (Fig. 4). Such a dependence disagrees with the usual quasiparticle picture, which would predict an inverse Hall angle $\frac{\sigma_{xx}}{\sigma_{xy}} \propto \frac{1}{\tau_{tr}} \propto T$ based on $\sigma_{xx} \propto \tau_{tr} \propto \frac{1}{T}$ and $\sigma_{xy} \propto (\tau_{tr})^2 \propto \frac{1}{T^2}$ [see e.g. Phys. Rev. Lett. 67, 2088 (1991)]. We note that the T-linear resistivity and T-square inverse Hall angle have been reported to support the strange metal phase in cuprates.

Third, the T-linear behavior only occurs in the vicinity of doping at which the Fermi liquid behavior ($\rho_{xx} \propto T^2$) disappears in the phase diagram; it does not occur over a wide range of doping density which would have been predicted by electron-phonon coupling [Phys. Rev. B 99, 165112 (2019)]. We therefore believe that there is strong enough experimental evidence showing strange metal behavior in our experiment.

In the revised manuscript (page 5, 6 and 10), we have included additional discussions on the various experimental manifestations of the strange metal phase. We have also included the estimate of T_{BG} and its comparison with the observed strange metal phase in the revised Methods.

6. In particular, you have not compared the observed T^* to the Debye temperature explicitly. Could you include this comparison or justify its omission?

Response 13:

We thank the reviewer for this question. The doping dependence of T^* (now we denote as T') cannot be explained by the doping dependence of T_{BG} . First, T' is present only over a narrow doping range of $(0.88 - 0.95)n_M$, where $T_{BG} \approx 17.5\text{K}$ is nearly a constant. (Note that the T' region is outside the small Hall density region of the phase diagram.) T_{BG} should be present over a wide doping range. Second, $T' (< 2\text{K})$ is substantially smaller than T_{BG} . Third, ρ_{xx} is expected to become super-linear in T for $T \lesssim T_{BG}$ rather than sub-linear in T as observed in our experiment. We are not aware of any electron-phonon theories that could predict a sub-linear T-dependence at low temperatures. Therefore, we believe that T' is not related to T_{BG} . We have included the above discussion in the revised Methods.

7. Given that direct spectroscopic evidence of a pseudogap is absent, might it be more appropriate to describe the observed behavior as a sub-linear resistivity region rather than a "possible pseudogap" phase?

Response 14:

We thank the reviewer for the suggestion. We have now removed the assignment of the phase to a pseudogap phase; we only compared the unusual temperature dependence in the T' region to a very similar dependence in the pseudogap phase of the cuprates on page 5.

8. In Line 221, you mention the system "possibly" exhibiting Planckian dissipation. Could you clarify whether your measured proportionality constant α (~ 1) is an experimentally extracted value or a theoretical expectation? Additionally, providing more precise numerical values and error bars

for α would strengthen your claims. For that matter, when I plug in the moiré density, their effective mass, and T-linear slope I obtain a value for alpha between 7 to 9, not 1. Please explain the choices you made when evaluating this figure more clearly.

Response 15:

We thank the reviewer for the comment. The proportionality constant $\alpha = \frac{\hbar}{\tau k_B T}$ (~ 1) is an order of magnitude estimate, not a measured value. Note a typo in the original manuscript; h should be \hbar [see e.g. Nature 595, 667–672 (2021)]. We used the measured $\frac{\Delta\rho_{xx}}{\Delta T} \approx 100\Omega/K$, calibrated moiré density $n_M \approx 6.7 \times 10^{16} \text{ m}^{-2}$ and an estimated effective mass $m^* \approx 0.5 m_0$ (m_0 is the free electron mass) to obtain $\alpha = \frac{\hbar}{\tau k_B T} \approx \frac{\hbar^2 \pi n_M}{m^* k_B} \frac{2e^2}{h} \frac{\Delta\rho_{xx}}{\Delta T} \approx 2.9$. There are two issues with this simple estimate. First, the effective mass cannot be measured in the strange metal phase in our experiment and a measured value away from it was used (Extended Data Fig. 2), but m^* is expected to be strongly doping dependent in this region. Second, we have assumed a quasiparticle description in the approximation which may not be applicable for strange metals. Considering the reviewer's comment and a related comment from reviewer #3 and #4, we have decided to remove the discussion on the Planckian scattering in the revised Methods. To strengthen the assignment of the strange metal phase, we have included additional discussions on various experimental manifestations of the strange metal. See **Response 12** for details.

9. In discussing tightly bound Cooper pairs based on moiré period and coherence lengths, why not use the experimentally measured Hall density, especially since the presence of the AFM insulator significantly modifies the simple estimate?

Response 16:

We thank the reviewer for the suggestion. As suggested by the reviewer, we can use the ratio of the coherence length ξ to the average inter-particle distance extracted from the Hall density ($\sim \sqrt{n_H} \xi$) as a measure of the dimensionless size of the Cooper pairs. We determined the coherence length $\xi \approx 34 \text{ nm}$ from the out-of-plane magnetic field dependence (Extended Data Fig. 4); the Hall density is about $n_H \approx 0.023 n_M$ in the under-doped superconducting region (Extended Data Fig. 13). The dimensionless ratio is $\sqrt{n_H} \xi \approx 1.4$, i.e. the Cooper pair size is comparable to the average inter-particle distance.

In the high- T_c literature, one often uses the ratio of the coherence length to the lattice scale as a dimensionless measure of the size of the Cooper pairs [see, for example, Mourachkine, A. High-Temperature Superconductivity in Cuprates: The Nonlinear Mechanism and Tunneling Measurements (Kluwer Academic Publishers, 2002)]. For instance, this ratio is about 5 in cuprates, showing that the Cooper pair size is only a few times of the lattice constant, i.e. tightly bound. For

comparison, we also determined the ratio of the coherence length to the moiré period because the moiré period is the lattice scale in this triangular lattice Hubbard system. Both measures show that the Cooper pairs are tightly bound. We have included the new estimate based on the Hall density in the revised Methods (page 9).

10. Lines 140 and 170 report two different Néel temperatures. Could you clarify whether these figures correspond to different devices or measurement conditions?

Response 17:

We thank the reviewer for the comment. The MCD measurements and transport measurements were performed on different samples because the contact gate electrodes in the transport devices significantly distort the incident light polarization and do not allow us to perform the MCD measurements. We have clarified this point in the revised manuscript and the caption of Fig. 2.

Addressing these points would significantly enhance the clarity and impact of the manuscript. I believe that with these points addressed, it would be of interest to both the moiré materials and high-T_c communities, and to researchers interested in 2D materials and strongly correlated materials more generally.

Response 18:

We thank the reviewer for the suggestions that have helped to improve our manuscript. We hope that the questions/comments have been sufficiently addressed in the revised version of the manuscript.

Referee #3 and #4 (Remarks to the Author):

In this manuscript, Xia et al. present a detailed study of superconductivity in twisted WSe₂ (tWSe₂) bilayers, focusing on devices with a twist angle of 4.6°, and reveal a phase diagram bearing a striking resemblance to that of high-temperature cuprate superconductors. While superconductivity has previously been reported in tWSe₂ systems with twist angles of 3.6° and 5°, this work uncovers several novel and unreported features, including:

- (1) direct evidence of antiferromagnetic (AF) order via magneto-optical measurements;
- (2) T-linear ‘strange metal’ resistivity over extended temperature and doping regimes; and
- (3) a crossover from T-linear to T² resistivity that closely mimics the transport evolution observed in cuprates.

The authors also report a Planckian dissipation rate, a high ratio of T_c/T_F, and potential signatures of a pseudogap in transport, though I find these latter interpretations more speculative. Overall, this is a remarkable and inspiring experimental study. The authors make impressive use of multiple tuning parameters—twist angle, carrier density, and displacement field—and combine dc transport and magneto-optical probes to construct the complex phase diagrams of a highly tunable and intricate system. The emergence of cuprate-like phenomenology in tWSe₂ is remarkable and, in my view, constitutes a significant advance in the field of unconventional superconductivity. I believe the work meets the bar for publication in Nature. However, I do have some comments that I would like to invite the authors to address the following points prior to further consideration.

Major points:

- 1) (Lines 46–48): The authors make a broad claim about the relevance of tWSe₂ to the high-T_c problem. Can they elaborate: In what specific ways can tWSe₂ offer new insights that has not already been achieved (or not possible to achieve) in the extensive literature of cuprates and other high-T_c systems?

Response 19:

We thank the reviewer for the positive assessment of our work. We are also grateful for the detailed suggestions which helped us to improve our manuscript. We believe that our work offers new insights beyond the existing high-T_c literature because of the relatively simple electronic structure, highly tunable electronic properties, and relatively low energy scales in tWSe₂.

- **Relatively simple electronic structure**

Moiré WSe₂ has a much simpler electronic structure near the Fermi level than cuprates and many other high-T_c materials. It has been shown that tWSe₂ could be well described by the Hubbard model on a triangular lattice [e.g. Physical Review Research 2, 033087 (2020), Physical Review

B 104, 075150 (2021), Nature Communications 12, 642 (2021) and Physical Review B 111, 014507 (2025)]. Our observation of high- T_c phenomenology in tWSe2 suggests that a simple Hubbard model could capture the essential physics of high- T_c superconductors, which remains a debate in the high- T_c literature.

- **Highly tunable electronic properties**

The ability to access nearly the entire superconducting phase diagram in a single tWSe2 sample by gating removes many complications or extrinsic effects (such as variable disorders) that cannot be avoided in cuprates and other high- T_c materials which rely on synthesis of many samples to access the entire phase diagram. In addition to the temperature and doping phase diagram, we also examined how the phase diagram evolves with bandwidth tuned by an out-of-plane electric field (see **Response 28** for details). The results show that the highest T_c is achieved right next to the Mott transition, where the antiferromagnetic insulator is quenched by the electric field. Moreover, to address the reviewer's comment, we performed additional experiments to map out the ground state phase diagram as a function of twist angle which also tunes the electronic bandwidth (revised Fig. 1). The result further confirms that, rather than following the location of the van Hove singularity, the superconducting state closely follows the bandwidth-tuned Mott transition. These observations strongly suggest that the superconducting state has its root in electronic correlations rather than pure electron-phonon coupling (otherwise, the most robust superconducting state would appear at the van Hove singularity).

- **Relatively low energy scales**

The low energy scales of moiré systems allow us to access the physics of higher effective energy scales which is impossible or difficult to access in high- T_c materials. This helps to shed new light on the parent states of the superconductors. For instance, we were able to thermally quench the antiferromagnetic insulator because of the low Mott/antiferromagnetic scale in tWSe2 (Fig. 3). We were also able to quench superconductivity using a small magnetic field (< 0.2 T) because of the low absolute T_c (although high relative T_c) in tWSe2 (Extended Data Fig. 4). By contrast, superconductivity can be quenched in high- T_c materials only by a very high magnetic field which often induces other phase transitions or complications. Particularly, the low upper critical field allowed us to access the strange metal phase down to the lowest possible temperature. As shown in our revised Fig. 4, we can now verify that the strange metal phase (after suppression of superconductivity) exists over a range of doping densities rather than a single critical density (or critical point). This is another open question in the high- T_c literature.

While it is hard to bring out all these points in the revised manuscript due to the length limit, we try to emphasize some of the points above in the revised manuscript (changes are highlighted).

2) Figure 2c: The Hall resistivity ρ_{xy} is positive in the electron-doped region and negative in the hole-doped region. This is opposite to conventional expectations, where electron-like carriers yield negative ρ_{xy} and hole-like carriers yield positive ρ_{xy} . The authors should clarify the sign convention used for ρ_{xy} in the Methods, or consider using the Hall coefficient R_H instead to avoid confusion.

Response 20:

We thank the reviewer for pointing this out. Following the suggestion, we have now used the convention for the sign of ρ_{xy} : negative for electrons and positive for holes. Note that the Hall density in Fig. 3i-l remains unchanged as we define electron (hole) density to be positive (negative) following the convention in the literature.

3) Figure 2e vs Figure 4a: For $E = -102$ mV/nm, $T_N \approx 8$ K (Fig. 2e), while for $E = -103$ mV/nm, $T_N \approx 3.5$ K (Fig. 4a). This is a surprisingly large discrepancy under nominally similar conditions. Can the authors clarify this inconsistency?

Response 21:

We thank the reviewer for the comment. The MCD and transport measurements were performed on different samples because the contact gate electrodes in the transport devices significantly distort the incident light polarization. We have not found a solution to this problem yet. The two measurements were performed in two different samples with 0.1-0.2 degrees difference in the twist angle (and under slightly different electric fields). There are also some sample-to-sample variations. These factors combined give rise to the observed difference in the Néel temperature. We have clarified this point in the revised manuscript and the revised caption of Fig. 2.

4) Figures 3i-k: The caption of Fig. 2 states that ρ_{xy} is inaccessible near $\nu = 1$ due to large ρ_{xx} in the insulating AF region, yet in Figs. 3i-k, Hall density appears to be inferred in that same region. Please clarify how Hall density is extracted in the AF regime.

Response 22:

We thank the reviewer for catching this. The Hall resistivity ρ_{xy} is indeed unreliable in the shaded region in Fig. 2c due to the large ρ_{xx} . To be consistent with Fig. 2c, we have now also shaded the same doping window in Fig. 3i-k to denote the regions with unreliable Hall density.

5) (Lines 201–204): The inference of a Planckian scattering rate is problematic. The authors use the Drude formula to extract τ , which requires knowledge of carrier density (n_M), effective mass

(m^*), and resistivity slope ($d\rho_{xx}/dT$). However, if I understand correctly, $d\rho_{xx}/dT$ is taken at $\nu = 0.88$ ($E = -103$ mV/nm), while m^* is measured at $\nu = 1.316$ from quantum oscillations. As shown in Fig. 3i, the A_2 coefficient (linked to m^*) varies significantly with ν . Therefore, using values from different ν to extract τ is questionable and may not be meaningful.

Response 23:

We agree with the reviewer. We could not determine m^* from quantum oscillations for fillings very close to the strange metal phase. Moreover, we observed clear quantum oscillations under magnetic fields above about 1-2 T, for which the spin-valley Zeeman effect may have already modified m^* from its zero-field value substantially. Given these constraints, it is difficult to accurately estimate the dimensionless parameter α for Planckian scattering. Considering this, we have removed the section on Planckian scattering in the revised Methods.

6) (Line 216): Similarly, the reported estimate of T_c/T_F at $E = -103$ mV/nm is likely not meaningful unless both T_c and T_F are determined at the same (or nearby) filling. Since both quantities are highly filling-dependent, comparing them at distinct ν undermines the validity of this ratio. I'd suggest removing this discussion unless the authors can extract m^* at or near the relevant filling.

Response 24:

We thank the reviewer for the comment. We have improved our estimate of $\frac{T_c}{T_F}$ in the under-doped region. We estimate $\frac{T_c}{T_F}$ for the superconductor denoted by a blue circle in the phase diagram (Fig. R5c, $\nu \approx 1.023$, $E = -99$ mV/nm), at which the superconducting dome intersects the antiferromagnetic metal (Fig. R5a). Figure R5b shows the T -dependence of ρ_{xx} and the Hall coefficient R_H . We observe an antiferromagnetic phase transition at $T_N \approx 1$ K, below which both ρ_{xx} and R_H increase due to Fermi surface reconstruction, followed by a superconducting phase transition at $T_c \approx 0.3$ K. The result confirms that the superconducting state develops from a small Fermi surface with density $(\nu - 1)n_M \approx 0.023 n_M$ ($n_M \approx 6.7 \times 10^{12} \text{cm}^{-2}$). Next, we extract the effective mass from the T -dependent Shubnikov-de Haas oscillations (Fig. R5e). However, clear quantum oscillations cannot be observed at a relatively low magnetic field at $E = -99$ mV/nm. At the same filling, quantum oscillations emerge when the electric field is tuned slightly away from $E = -99$ mV/nm. This indicates that the effective mass decreases. Therefore, we used the measured effective mass ($m^* \approx 0.5 m_0$) at $E = -105$ mV/nm (denoted by a pink circle in Fig. R5c) as an estimate of the lower bound of the effective mass. This allows us to estimate $T_F \approx \frac{\hbar^2 \pi * 0.023 n_M}{m^* k_B} \approx 8.5$ K and obtain $\frac{T_c}{T_F} \approx 3.5\%$ as a lower bound [the value is on par with our earlier work, Nature 637, 833–838 (2025), in which T_F can be directly obtained as the coherence temperature for electrical transport]. Nevertheless, the estimate shows a high $\frac{T_c}{T_F}$ ratio, which is also

consistent with the observed high $\frac{T_c}{T_N}$ ratio (Extended Data Fig. 11). We have provided this more accurate estimate in the revised Methods (page 9) and included Fig. R5 in Extended Data Fig. 2 and 13.

Figure R5. Estimation of T_c/T_F . **a**, ρ_{xx} versus ν and T at $E = -99$ mV/nm and $B = 0$ T. **b**, Temperature dependence of ρ_{xx} (blue) and Hall coefficient R_H (red) at $\nu = 1.023$, where the superconducting dome in **a** overlaps with the AF metal. An AF metal with a small Fermi surface is observed below T_N , manifested by a bump in ρ_{xx} and a jump in R_H . The AF metal turns into a superconductor at $T_c \approx 280$ mK. The result demonstrates that superconductivity in the underdoped region is developed from an AF metal with a small Fermi surface. **c**, ρ_{xx} versus ν and E at $T = 50$ mK and $B = 0$ T. The blue (pink) circle marks where panel **b** (the effective mass m^*) was measured. **d**, ρ_{xx} versus ν and B measured along the black dashed line in **c**. A Landau fan emerges from $\nu = 1$, showing a small Fermi surface with carrier density $(\nu - 1)n_M$. **e**, Resistivity difference $\Delta\rho_{xx} = \rho_{xx}(T) - \rho_{xx}(1.4\text{ K})$ as a function of ν at varying T and fixed $B = 2$ T (dashed line in **d**). **f**, T -dependence of the amplitude $|\Delta\rho_{xx}|$ at $\nu \approx 1.023$ as marked by the arrow in **e**. $|\Delta\rho_{xx}|$ was obtained as the difference between the nearest $\Delta\rho_{xx}$ peak and dip centered around ν . The quasiparticle effective mass m^* was extracted by fitting the data to $|\Delta\rho_{xx}| = \frac{R_a\lambda(T)}{\sinh\lambda(T)}$ (red lines). Here R_a is the amplitude and $\lambda(T) = \frac{2\pi^2 k_B T m^*}{\hbar e B}$.

Minor points (optional):

1) Fig. 1b: Please mark the position of the van Hove singularity (vHs) for $E = 0$ explicitly.

Response 25:

Position of the vHS is now marked.

2) Figs. 1c&d: The authors should clarify whether a higher density of states necessarily leads to higher resistance in tWSe₂. This is not obviously expected and merits a brief explanation.

Response 26:

We thank the reviewer for the suggestion. In the simple Drude picture, 2D resistivity, $\rho_{xx} = \frac{N(\epsilon_F) \hbar}{n} \frac{h}{\tau 2e^2}$, is determined by the electronic density of states at the Fermi energy $N(\epsilon_F)$, the 2D doping density n , and the transport scattering rate $1/\tau$. In general, a higher density of states implies a higher resistivity. In addition, a higher density of states can also enhance the electron scattering rate $1/\tau$ (e.g. arxiv:2503.11763). Both factors enhance ρ_{xx} . We have added a brief explanation in the revised manuscript on page 2.

3) Figs. 2e&f: Consider showing a Curie-Weiss analysis for the magnetic susceptibility, particularly to support the AF regime, with an explicit negative T_{CW} if applicable.

Response 27:

We thank the reviewer for the suggestion. We have now included the Curie-Weiss analyses for the data in Fig. 2f and the doping dependence of the Curie-Weiss temperature in Extended Data Fig. 5. The result clearly shows that the Curie-Weiss temperature is negative for the antiferromagnetic states and its magnitude is comparable to the measured Néel temperature.

4) Could the authors clarify the role of the displacement field in shaping the phase diagram of tWSe₂? For instance, is E used to bring the system closer to a correlated insulator near $\nu = 1$? An explicit illustration for this would make the manuscript easier to follow for a non-expert.

Response 28:

As shown in the phase diagram in Fig. 1b,c, the location of the van Hove singularity (vHS) is a function of (E, ν) . For a given ν (for instance, half-band-filling at $\nu = 1$), the electric field tunes the location of the vHS relative to the Fermi level and, therefore, also the Fermi velocity. Specifically, the closer the vHS is to the Fermi level, the lower is the Fermi velocity. In other words, the electric field tunes the flatness of the bands at the Fermi level. Flatter bands correspond to stronger correlations and favor a correlated insulator at $\nu = 1$. This explains the absence of a correlated insulator at $\nu = 1$ for both small and large electric fields in samples with intermediate

twist angles ($\theta \approx 3.6^\circ - 4.6^\circ$), where the vHS is far away from the Fermi level. Only for intermediate electric fields, the vHS is close to the Fermi level and the reduced Fermi velocity stabilizes a correlated insulator. For $\theta \gtrsim 4.7^\circ$, the correlations are too weak to stabilize the insulator even with the help of the vHS. And for $\theta \lesssim 3.5^\circ$, the insulator is always stabilized in the layer-hybridized region due to the narrow moiré bandwidth regardless of the location of the vHS. Effectively, both the twist angle and the electric field can tune the band flatness at the Fermi level and realize a bandwidth-tuned Mott transition at $\nu = 1$. We have included the above discussion in the revised Methods on page 8.

Referee #1 (Remarks to the Author):

The authors have satisfactorily addressed all of my questions and resolved the concerns I previously had regarding this work. After reviewing the revised manuscript and the authors' responses to the other two referees, I now support the publication of this manuscript in Nature.

Optional further question:

The data presented in Figure 1(e–g) of the revised manuscript are quite compelling. In Response 3, the authors state: “We expect T_c to drop quickly at large twist angles because the moiré effects are washed out and WSe₂ without moiré effects is not a superconductor under our experimental conditions.” This raises a natural question: Have the authors performed measurements on samples with larger twist angles, specifically beyond 5 deg? If not, is there a particular reason—such as fabrication limitations or sample quality—that restricts the experimental exploration to twist angles below this threshold?

Response 1:

We thank the reviewer for the additional question. In this study, we focus on superconductivity and the complex phase diagram near the Mott transition. We have not measured samples of twist angle $> 5^\circ$ down to the dilution temperature. Future studies will systematically vary the twist angle in the limit of large angle and determine the value beyond which superconductivity ceases to exist.

Referee #2 (Remarks to the Author):

The authors have largely addressed my concerns. In my opinion, the additions and modifications to the manuscript have significantly improved it. The twist-angle dependence of the low-temperature phase diagram in particular is a valuable addition and strengthens their argument that this system enables bandwidth tuning not available in high-T_c systems.

I have only one major concern remaining (more of a recommendation perhaps) and two minor ones. My major concern is the title. I still do not feel it is appropriate to describe this system as a "simulation" of high-temperature superconductors. As the authors themselves note in their rebuttal, if the work simulates anything, it is the triangular-lattice Hubbard model. They can tune the bandwidth and investigate how this modifies the phase diagram. I agree that this is a compelling contribution to the study of Hubbard lattices and that it may shed light on the behavior of more complicated systems such as high-temperature superconductors. Considering this, I think the title should reflect the true novelty of their work and avoid overstating (even unintentionally) the connection to high-temperature superconductors. For instance, titles like "Programmable triangular-lattice Hubbard physics in twisted WSe₂", or "Bandwidth-tuned Mott transition and superconductivity in twisted WSe₂" would keep the focus where it belongs, on the novel tunability afforded by twist angle and electric-field control of the bandwidth.

Response 2:

We thank the reviewer for thoughtful suggestions. We have changed the title to "Bandwidth-tuned Mott transition and superconductivity in moiré WSe₂" as suggested.

My two minor concerns are as follows:

1. The revised Fig. 1 is significantly improved over the original by the lower-temperature resistance maps and twist-angle analysis, but the twist angles do not exactly match those in the original Fig. 1. Did the twist-angle estimates for each of these devices change, or are these measurements from new devices?

Response 3:

We thank the reviewer for careful reading of our manuscript. There is no change in the twist angle determination. Results from new devices are included in the revised Fig. 1.

2. The "overall trend" of T_{BKT} versus twist suggested by the shaded region and the dashed line seems insufficiently justified. The data appear to show a cluster around 200 mK, two devices near 350 mK, and two that do not superconduct. That information is valuable on its own, but the trend is not clear enough to highlight based on these points alone. Otherwise, are the authors indicating that they expect T_{BKT} to continue increasing beyond 5 degrees, as the trend line implies?

Overall, I find the revisions thorough and the results compelling. I believe the results are high impact and will be of broad interest, and are thus suitable for publication in Nature.

Response 4:

We thank the reviewer for the comment. Although we are confident in the overall trend of the optimal T_{BKT} versus twist angle, we agree that the dashed line with shaded band is not sufficiently justified based on the available experimental data. The trend is further complicated by the sensitivity of T_{BKT} to sample quality. In addition, we expect T_{BKT} to decrease rapidly with increasing twist angle beyond $\sim 5^\circ$, but further experiment is required to determine the exact twist angle (**Response 1**). In the revised Fig. 1g, we have removed the line; we have also included an additional data point a 5-degree device (Ref. 8), which helps to further illustrate the overall trend.

Referee #3 & #4 (Remarks to the Author):

In the revised version the authors have taken considerable efforts to address the comments raised by myself and other reviewers. In my view the manuscript has now improved in consistency and clarity, and I can recommend its publication in its current form. I have a few minor comments that I would like to invite the authors to address before finalizing the manuscript:

- Throughout the manuscript the authors used the term “high-Tc superconductivity” to describe the superconductivity in the cuprates. However, there are other distinct materials systems that are considered as high-Tc materials but do not share the same phase diagram as the cuprates, such as the iron-based superconductors, the hydrides, and the nickelates. I’d suggest the authors to use an alternative term e.g. cuprate superconductivity, to clarify that the “simulation of high-Tc superconductivity” in the tWSe₂ system refers specifically to the cuprates.

Response 5:

We thank the reviewer for the suggestion. We have clarified in the revised manuscript that our comparison is made specifically with the cuprate superconductors.

- Lines 235-237: here the authors state that the Hall density jump for the $E=-87$ mV/nm case without AFM is about $2 \cdot n_M$ (Fig. 3l), different from the jump of $\sim n_M$ found in the cases where AFM is present (Fig. 3i-k). However, to my eyes, the n_H jumps do not look different across Fig. 3i,j,k,l. Can the authors clarify on what they mean by the difference in the n_H jump size here?

Response 6:

We thank the reviewer for the comment. The three lines in Fig. 3i-l illustrate hole filling dependences of the Hall density n_H in units of moiré density: $2 - \nu$, $1 - \nu$, and $-\nu$. In Fig. 3i, the Hall density follows $1 - \nu$ in the vicinity of $\nu = 1$, consistent with a small Fermi surface for an antiferromagnetic metal (note that the Hall density cannot be measured in the shaded region for the correlated insulator centered at $\nu = 1$). The Hall density jumps at the peak of the two superconducting domes and follows $2 - \nu$ for $\nu > 1$ and $-\nu$ for $\nu < 1$ away from the jumps. The step of each jump is one moiré density. The jumps arise from transitions from the antiferromagnetic metal to an electron- (hole-like) Fermi liquid with large Fermi surface for $\nu > 1$ ($\nu < 1$). Similar behaviors are observed for two other displacement fields in Fig. 3j and 3k. By contrast, in Fig. 3l, the small Fermi surface state is absent; there is only one jump in the Hall density across a van-Hove singularity near $\nu = 1$; the Hall density follows $2 - \nu$ for $\nu > 1$ (an electron-like Fermi liquid) and $-\nu$ for $\nu < 1$ (a hole-like Fermi liquid) away from the jump. The step of the jump is two times moiré density. We have expanded the discussion of Fig. 3i-l in Methods to clarify the difference in the Hall density jump for different regions of the phase diagram.

- Formatting: in the figures all the parameters and units are correctly labelled; in the text, however, there’s no space between the value and units, and the subscripts are styled differently than in the figures. The authors should revise the text for correctness and consistency with the figures.

Response 7:

We thank the reviewer for careful reading of our manuscript. We have corrected the format of the variables and their values and units in the revised text.